# Impact of maternal antibodies and microbiota development on the immunogenicity of oral rotavirus vaccine in African, Indian, and European infants

Edward P. K. Parker [1,11✉], Christina Bronowski [2,11], Kulandaipalayam Natarajan C. Sindhu [3,11], Sudhir Babji[3], Blossom Benny[3], Noelia Carmona-Vicente[2], Nedson Chasweka[4], End Chinyama[4], Nigel A. Cunliffe[2,5], Queen Dube[4], Sidhartha Giri[3], Nicholas C. Grassly [6], Annai Gunasekaran[3], Deborah Howarth[2], Sushil Immanuel[3], Khuzwayo C. Jere [2,4,7], Beate Kampmann [1], Jenna Lowe[2], Jonathan Mandolo[4], Ira Praharaj[3], Bakthavatsalam Sandya Rani[3], Sophia Silas[3], Vivek Kumar Srinivasan[3], Mark Turner [8], Srinivasan Venugopal[3], Valsan Philip Verghese[9], Alistair C. Darby[2,12], Gagandeep Kang[3,12] & Miren Iturriza-Gómara [2,10,12✉]

Identifying risk factors for impaired oral rotavirus vaccine (ORV) efficacy in low-income countries may lead to improvements in vaccine design and delivery. In this prospective cohort study, we measure maternal rotavirus antibodies, environmental enteric dysfunction (EED), and bacterial gut microbiota development among infants receiving two doses of Rotarix in India (n = 307), Malawi (n = 119), and the UK (n = 60), using standardised methods across cohorts. We observe ORV shedding and seroconversion rates to be significantly lower in Malawi and India than the UK. Maternal rotavirus-specific antibodies in serum and breastmilk are negatively correlated with ORV response in India and Malawi, mediated partly by a reduction in ORV shedding. In the UK, ORV shedding is not inhibited despite comparable maternal antibody levels to the other cohorts. In both India and Malawi, increased microbiota diversity is negatively correlated with ORV immunogenicity, suggesting that high early-life microbial exposure may contribute to impaired vaccine efficacy.

[1] The Vaccine Centre, Department of Clinical Research, London School of Hygiene and Tropical Medicine, London WC1E 7HT, UK. [2] Institute of Infection, Veterinary and Ecological Sciences, University of Liverpool, Liverpool L69 7BE, UK. [3] Division of Gastrointestinal Sciences, Christian Medical College, Vellore, Tamil Nadu 632004, India. [4] Malawi-Liverpool-Wellcome Trust Clinical Research Programme, University of Malawi, BlantyrePO Box, 30096, Malawi. [5] NIHR Health Protection Research Unit in Gastrointestinal Infections, University of Liverpool, Liverpool, UK. [6] Department of Infectious Disease Epidemiology, Imperial College London, London W2 1PG, UK. [7] Department of Medical Laboratory Sciences, College of Medicine, University of Malawi, Private Bag 360, Chichiri, Blantyre 3, Malawi. [8] Institute of Life Course and Medical Sciences, University of Liverpool, Liverpool L8 7SS, UK. [9] Department of Child Health, Christian Medical College, Vellore, Tamil Nadu 632004, India. [10] Centre for Vaccine Innovation and Access, PATH, Geneva, Switzerland. [11] These authors contributed equally: Edward P. K. Parker, Christina Bronowski, Kulandaipalayam Natarajan C. Sindhu. [12] These authors jointly supervised this work: Alistair C. Darby, Gagandeep Kang, Miren Iturriza-Gómara. ✉email: edward.parker@lshtm.ac.uk; miturrizagomara@path.org

The roll-out of oral rotavirus vaccine (ORV) has now reached over 100 countries. This initiative—in parallel with advances in sanitation infrastructure and increased use of oral rehydration therapy—has led to substantial declines in global diarrhoeal mortality[1]. Whereas rotavirus was linked with over 500,000 infant deaths annually at the turn of the century[2], this number currently stands at approximately 130,000[3]. Yet the potential impact of ORV is constrained by the impaired performance of current vaccines in low- and middle-income countries (LMICs). The 1-year protective efficacy of Rotarix against severe rotavirus-associated gastroenteritis is >95% in Europe[4] but may fall below 50% in sub-Saharan Africa[5]. Moreover, while a variety of interventions to boost ORV performance have been tested (e.g. temporary withholding of breastfeeding), these have generally proven either ineffective or of modest benefit[6], highlighting the need for new strategies informed by a deeper understanding of the mechanisms underlying the vaccine efficacy gap.

A variety of risk factors have been linked with impaired oral vaccine performance in LMICs[7]. Passively acquired maternal antibodies appear to interfere with ORV immunogenicity[8], as documented for a variety of parenteral vaccines[9]. However, it remains unclear whether rotavirus-specific antibody concentrations and their inhibitory effect are higher in LMICs. Environmental enteric dysfunction (EED)—a subclinical condition associated with intestinal inflammation and increased permeability—is common in LMICs and has been cited as a possible cause of oral vaccine failure[10]. However, EED is generally measured using faecal or plasma markers as a proxy for mucosal immune status, and studies of these markers at the time of oral vaccine delivery have yielded mixed results[11]. The bacterial gut microbiota shapes and is in turn shaped by the developing infant immune system. Among infants in Ghana, Rotarix immunogenicity was positively correlated with the relative abundance of *Streptococcus bovis* and negatively correlated with Bacteroidetes abundance at the time of the first vaccine dose[12]. However, no significant discrepancies in microbiota composition were apparent when comparing Rotarix responders with non-responders in India[13]. Consistent predictors of ORV failure within the bacterial gut microbiota thus remain elusive.

Much of what we know about the mechanisms shaping oral vaccine performance comes from single-population studies focusing on individual risk factors. Here, we present a multicentre cohort study exploring the effect of maternal antibodies, EED markers, and bacterial gut microbiota development on Rotarix response among infants in Malawi, India, and the UK.

## Results

**Study population.** We enrolled pregnant women during the third trimester in Vellore (India; $n = 395$), Blantyre (Malawi; $n = 187$), and Liverpool (UK; $n = 82$). After delivery, infants received routine vaccines including two doses of Rotarix according to the national immunisation schedule (weeks of life 6 and 10 in India and Malawi; weeks of life 8 and 12 in the UK). We measured rotavirus-specific IgA (RV-IgA) in maternal blood, cord blood, and breastmilk samples collected during or in the week after delivery, and in infant blood samples collected pre- and post-vaccination (4 weeks after dose 2). In Indian participants, all serum samples were also assayed for rotavirus-specific IgG (RV-IgG). Starting in the first week of life, six longitudinal stool samples were collected from each infant and assayed for rotavirus shedding, with samples collected 1 week after each ORV dose providing an indicator of vaccine virus take. As a proxy for bacterial microbiota development in the infant gut, we sequenced the V3–V4 region of the 16S rRNA gene in stool samples collected at 1 and 4 weeks of age, and before each ORV dose. EED

markers were measured in serum and/or stool samples collected before each vaccine dose (Fig. 1A).

A total of 486 infants (307 in India, 119 in Malawi, and 60 UK), born between December 2015 and April 2018, met the primary endpoint for inclusion in the analysis (measurement of seroconversion or dose 1 shedding). While the pre- and post-parturition conditions of these disparate cohorts are innumerable, several distinguishing features are highlighted in Table 1. Infants in the UK were characterised by a higher birthweight and a greater prevalence of formula feeding (though >75% were partially or exclusively breastfed). Elective caesarean was an exclusion criterion for the UK but was the mode of delivery for 70/307 (23%) infants in India. HIV exposure was common among infants in Malawi (27/119 [23%]).

Other vaccines were administered according to the routine schedule at each site, including OPV at 0, 6, and 10 weeks of age in Malawi and India. In India, 98/307 (32%) infants were born before April 2016—the date of the global switch from tOPV to bOPV. Infants therefore received tOPV-only, bOPV-only, or mixed schedules (Table 1). To explore the potential inhibitory effect of OPV on ORV[14], the 6 and 10 week doses of OPV were replaced with inactivated poliovirus vaccine (IPV) in a sequentially recruited cohort in India (100/307 [33%]). The comparison of OPV and IPV arms is described elsewhere (Babji et al., in preparation). Briefly, while shedding after the first dose of ORV was less common in OPV than IPV recipients, cumulative shedding after both doses was comparable in the two arms, as were seroconversion rates and post-vaccination RV-IgA levels. Given the lack of association between study arm and ORV immunogenicity, we pooled the IPV and OPV arms in the present study.

**ORV shedding and immunogenicity.** In the UK, we observed near-ubiquitous shedding of the first dose of ORV, with 55/60 (92%) shedding 1 week after vaccination and 24/54 (44%) continuing to shed immediately prior to the second dose (4 weeks after dose 1; Supplementary Fig. 1A). By contrast, dose 1 shedding was detected in 82/305 (27%) infants in India and 56/101 (55%) in Malawi (Fig. 1B), and continued shedding prior to the second dose was much rarer in these cohorts (Supplementary Fig. 1A). Shedding following at least one dose was observed in 54/56 (96%) infants in the UK, 151/304 (50%) in India, and 50/72 (69%) in Malawi (Fisher's p values <0.005 for all between-country comparisons after false discovery rate [FDR] adjustment; Fig. 1B).

We observed similar geographic discrepancies in ORV immunogenicity. Baseline seropositivity was common in India (99/305 [32%] compared to <5% in the UK and Malawi) due to high rates of neonatal rotavirus infection (see below). Seroconversion, defined as detection of RV-IgA at ≥20 IU/ml in previously seronegative infants or a 4-fold increase in RV-IgA concentration among infants who were seropositive at baseline, was observed in 85/305 (28%) infants in India, 24/103 (23%) infants in Malawi, and 27/51 (53%) infants in the UK (Fig. 1C). Geometric mean concentrations (GMCs) of RV-IgA (IU/ml) after vaccination were 20 (95% CI 16–25) in India, 9 (6–12) in Malawi, and 27 (17–45) in the UK (Tukey's post-hoc p values <0.005 for comparisons between Malawi and other cohorts and 0.489 for India vs UK). Among infants who seroconverted, post-vaccination RV-IgA levels did not differ significantly among cohorts (GMCs of 93 [73–118], 122 [80–187], and 105 [71–155] in India, Malawi, and the UK, respectively; Tukey's post-hoc p values >0.05).

**Neonatal rotavirus infection.** A distinct feature among Indian infants was the high rate of neonatal rotavirus infection, which we

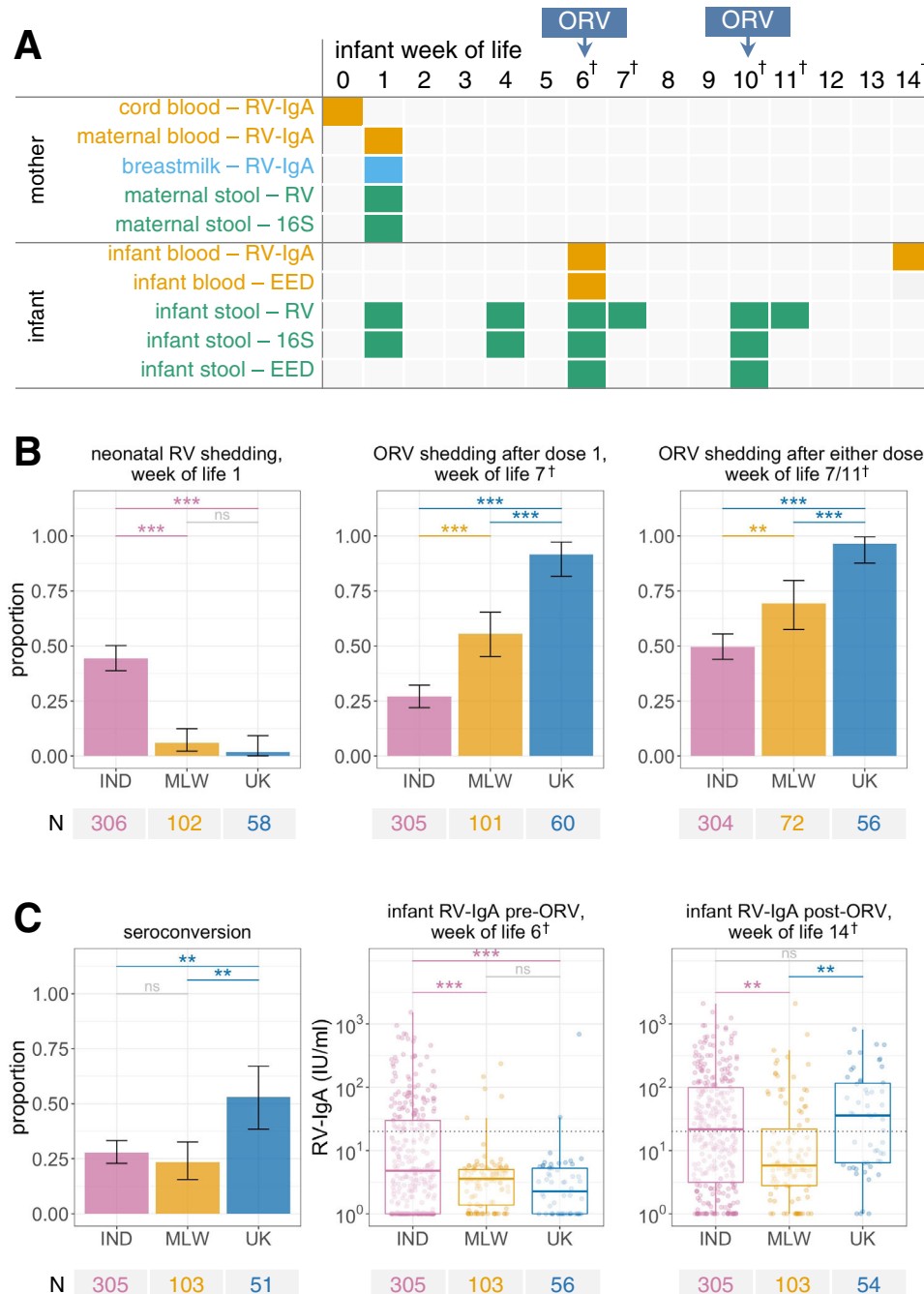

**Fig. 1 Study design and oral rotavirus vaccine response. A** Study design. The final study population comprised 307 infants in India, 119 in Malawi, and 60 in the UK. **B**, **C** Geographic differences in **B** rotavirus shedding and **C** ORV immunogenicity. Rotavirus shedding was detected via quantitative PCR using a pan-rotavirus assay targeting the *VP6* gene of group A rotaviruses (week of life 1) and an assay for vaccine virus shedding targeting the Rotarix *NSP2* gene (1 week after each dose). Seroconversion was defined as detection of RV-IgA at ≥20 IU/ml post-vaccination among infants who were seronegative at baseline or a 4-fold increase in RV-IgA concentration among infants who were seropositive at baseline. Error bars represent Clopper–Pearson 95% confidence intervals. Groups were compared by two-sided Fisher's exact test with FDR correction (binary outcomes) or ANOVA with post-hoc Tukey tests (continuous outcomes). The dotted lines at 20 IU/ml indicate the standard cut-off for RV-IgA seropositivity. Box plots display median (centre line), upper and lower quartiles (box limits), the minimum value greater than or equal to the lower quartile − 1.5 × interquartile range (lower whisker), and the largest value less than or equal to the upper quartile + 1.5 × interquartile range (upper whisker). EED, environmental enteric dysfunction markers; IND, India; MLW, Malawi; ns, not significant; ORV, oral rotavirus vaccine; RV, rotavirus; †, +2 weeks in UK due to later vaccination schedule; *$p < 0.05$; **$p < 0.005$; ***$p < 0.0005$.

defined as the detection of wild-type rotavirus shedding in week of life 1 (Fig. 1B) and/or baseline seropositivity (pre-vaccination RV-IgA ≥20 IU/ml). This was observed in 166/304 (55%) infants in India, whereas the corresponding rates were 10/90 (11%) in Malawi and 2/54 (4%) in the UK. Neonatal infection in India was

more common among infants born in tertiary care facilities (relative risk [RR] 1.98 [95% CI 1.72–2.19]; Supplementary Table 1) and among infants delivered by caesarean section versus vaginal delivery (RR 1.31 [1.05–1.53]). All neonatal rotavirus infections were asymptomatic. We successfully characterised

**Table 1 Baseline characteristics of study cohorts.**

|  | India | Malawi | UK |
|---|---|---|---|
| N | 307 | 119 | 60 |
| Month of birth |  |  |  |
| First | 12/2015 | 11/2016 | 09/2016 |
| Last | 11/2016 | 04/2018 | 03/2018 |
| Caesarean delivery | 70 (22.8) | 0 (0.0) | 1 (1.7)[a] |
| Birthweight (kg) | 2.96 (0.42) | 3.02 (0.42) | 3.68 (0.48) |
| Female | 152 (49.5) | 60 (50.4) | 24 (40.0) |
| Polio vaccine schedule |  |  |  |
| tOPV | 57 (18.6) | — | — |
| Mixed tOPV/ bOPV | 56 (18.2) | — | — |
| bOPV | 94 (30.6) | 119 (100.0) | — |
| IPV | 100 (32.6) | — | 60 (100.0) |
| Breastfeeding[b] |  |  |  |
| Exclusive | 265 (86.3) | 108 (90.7) | 26 (43.3) |
| Partial | 32 (10.4) | 11 (9.2) | 20 (33.3) |
| None | 10 (3.3) | 0 (0.0) | 14 (23.3) |
| Exposed to antibiotics[c] | 84 (27.4) | 33 (27.7) | 6 (10.0) |
| HIV status |  |  |  |
| Exposed | 0 (0.0) | 27 (22.7) | — |
| Unexposed | 299 (97.4) | 88 (73.9) | — |
| Unknown | 8 (2.6) | 4 (3.4) | 60 (100.0) |

Data are n (%) or mean (s. d.).
[a]Elective caesarean was an exclusion criterion for this cohort (with 1 delivery by emergency caesarean).
[b]Data from birth to week of life 11 in Malawi and India, and birth to week of life 13 in the UK.
[c]Data from birth to week of life 14 in Malawi and India, and birth to week of life 13 in the UK.

rotavirus genotype in 104 samples with detectable rotavirus shedding at week of life 1, of which 103 were positive for the strain G10P[11].

Consistent with the high prevalence of neonatal rotavirus infection, we observed significantly higher pre-vaccination RV-IgA concentrations in India compared with Malawi and the UK (Fig. 1C). Neonatal infection was associated with a reduced likelihood of dose 1 ORV shedding (RR 0.47 [0.30–0.71]). By contrast, neonatal infection did not significantly influence the likelihood of seroconversion (RR 1.25 [0.86–1.73]). Where ORV shedding 1 week after either dose was observed among infants with neonatal infection, this significantly boosted post-vaccination RV-IgA concentrations, pointing to a cumulative effect of neonatal infection and vaccination on overall immunogenicity (Supplementary Fig. 1B). Indeed, while the post-vaccination GMCs of Indian infants lacking neonatal infection were commensurate with those observed in Malawi (6 [5–8] vs 9 [6–12], respectively), the final antibody levels among Indian infants with neonatal infection exceeded those observed in the UK (55 [42–73] vs 27 [17–45], respectively).

**Breastfeeding, growth, and sanitation.** We examined several baseline health and demographic variables for their potential correlation with dose 1 ORV shedding (Supplementary Data 1), seroconversion (Supplementary Data 2), and post-vaccination RV-IgA (Supplementary Data 3). In India, seroconversion was positively correlated with exclusive breastfeeding (RR 2.04 [1.07–3.68]) and was less common in infants who received tOPV-containing than bOPV-only schedules (RRs of 0.56 [0.29–0.95] for tOPV-only and 0.56 [0.29–0.95] for mixed tOPV/bOPV), although the potential contribution of seasonal changes (e.g., in enteropathogen exposure) to this trend cannot be discounted given that OPV schedule was not randomly allocated. Similar associations were apparent in India when considering post-

vaccination RV-IgA concentration as an endpoint (Supplementary Data 3). Moreover, RV-IgA levels were positively correlated with height-for-age Z score at the time of the first ORV dose in India (beta 1.32 [95% CI 1.07–1.63], where beta represents the estimated ratio of GMCs per unit change in Z score), and was higher among infants in India living in houses built from permanent versus temporary or mixed materials (beta 2.03 [1.26–3.26]) or with access to treated water (beta 1.66 [1.04–2.67])—both indicators of higher socioeconomic status. Baseline health and demographic variables were not significantly correlated with ORV response in Malawi or the UK, although fewer covariates were measured in these cohorts (Supplementary Data 1–3).

**Maternal antibodies.** Serum RV-IgA concentrations were significantly higher among mothers in Malawi than in India or the UK (Fig. 2A; GMCs of 134 [120–150], 340 [256–452], and 186 [123–281] in India, Malawi, and the UK, respectively). Interestingly, maternal serum RV-IgA levels did not differ significantly between India and the UK, while breastmilk RV-IgA concentrations were significantly higher in Malawi and the UK than in India (GMCs of 25 [22–29], 97 [84–112], and 100 [68–146] in India, Malawi, and the UK, respectively). Reflecting the relative deficit in breastmilk RV-IgA levels in India, maternal breastmilk/serum RV-IgA ratios were significantly lower in this cohort than both Malawi and the UK (Fig. 2B).

Our analysis of the association between maternal antibodies and ORV outcome yielded several notable findings. First, although maternal serum RV-IgA concentrations were not significantly associated with seroconversion (RRs of 0.94 [0.78–1.12], 0.80 [0.61–1.01], and 0.91 [0.70–1.09] in India, Malawi, and the UK, respectively; Supplementary Data 2), they were negatively correlated with infant post-vaccination RV-IgA levels in each cohort (Pearson coefficient [r] of −0.29, −0.13, and −0.12 in Malawi, India, and the UK, respectively; Fig. 2C). Although this correlation was not significant in the UK, we observed no significant evidence of effect modification between cohorts (p values of >0.05 for interaction terms between maternal RV-IgA and country). Second, maternal serum RV-IgA levels were negatively correlated with ORV shedding after dose 1 in India (RR 0.75 [0.60–0.92]), particularly among infants with neonatal infection (RR 0.50 [0.31–0.75]), and a similar trend was evident in Malawi (RR 0.85 [0.67–1.00]; Supplementary Data 2). By contrast, ORV shedding in the UK was ubiquitous despite the fact that maternal antibody levels were equivalent to those in India. Finally, we observed no significant difference in maternal antibody levels according to rotavirus shedding in week of life 1 among Indian infants (Supplementary Table 1 and Fig. 2D), suggesting that neonatal G10P[11] viruses may not be subject to the same shedding inhibition observed for ORV. However, among infants with neonatal infection (as defined above), we observed a strong negative correlation between maternal RV-IgG levels and infant RV-IgA at 6 weeks of age (r −0.42, p < 0.001; Fig. 2C).

Disentangling the relative influence of breastmilk versus transplacental antibodies is challenging given the correlation between the two (r of 0.29, 0.39, and 0.29 in India, Malawi, and the UK, respectively; Fig. 2D). As observed for maternal serum RV-IgA, breastmilk RV-IgA was not significantly correlated with seroconversion in any cohort (Supplementary Data 3), but was negatively correlated with infant post-vaccination RV-IgA levels in India and Malawi (r of −0.14 and −0.26, respectively; Fig. 2C). Breastmilk RV-IgA was negatively correlated with ORV shedding after dose 1 in India (RR 0.83 [0.69–0.98]; Supplementary Data 2) but not in Malawi (RR 0.91 [0.58–1.23]).

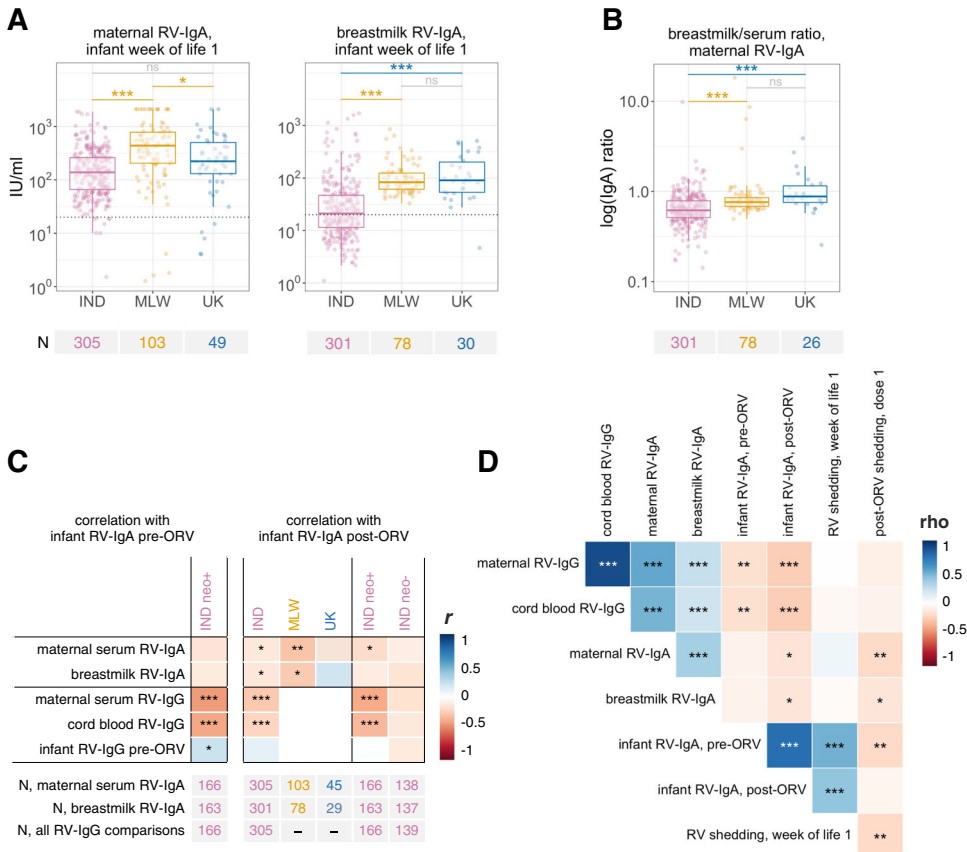

**Fig. 2 Association between maternal antibodies and oral rotavirus vaccine response. A** Geographic differences in maternal antibody concentrations. Groups were compared by ANOVA with post-hoc Tukey tests. The dotted lines at 20 IU/ml indicate the standard cut-off for RV-IgA seropositivity. **B** Geographic differences in maternal breastmilk/serum RV-IgA ratios. Groups were compared by Dunn's test. Ratios were calculated using log-transformed antibody concentrations. See Fig. 1 legend for box plot parameters. **C** Association between maternal antibodies and infant RV-IgA formation. Log-transformed concentrations were compared using Pearson's correlation coefficient (*r*) with two-sided hypothesis testing. Infant samples for RV-IgA measurement were collected at the time of dose 1 (week of life 6 in India/Malawi; week of life 8 in the UK) and 4 weeks after dose 2 (week of life 14 in India/Malawi; week of life 16 in the UK). **D** Correlation between rotavirus-specific antibody concentrations and rotavirus shedding in Indian infants with complete data ($n = 298$). For shedding variables, 1/Ct was used such that higher values correspond to higher rotavirus quantities. Shedding after week of life 1 was determined based on the group A rotavirus *VP6* gene assay (Ct range 23.5–35.0) while shedding after dose 1 was based on the Rotarix-specific *NSP2* gene assay (Ct range 20.7–40.0). Variables were compared using Spearman's rank correlation coefficient (rho) with two-sided hypothesis testing. neo+, infected with rotavirus neonatally (defined by detection of rotavirus shedding in week of life 1 or baseline seropositivity); neo−, uninfected with rotavirus neonatally; ns, not significant; ORV, oral rotavirus vaccine; RV, rotavirus; *$p < 0.05$; **$p < 0.005$; ***$p < 0.0005$.

**Inflammatory biomarkers**. We observed strong geographic discrepancies in EED markers. Whereas α1-antitrypsin (α1AT; a marker of protein-losing enteropathy) was highest in Malawi and lowest in the UK, both myeloperoxidase (MPO; a marker of neutrophil activity) and α1 acid glycoprotein (a marker of systemic inflammation) were highest in Indian infants (Supplementary Fig. 2A). Despite their marked variation within and between cohorts, none of the biomarkers were significantly associated with seroconversion, dose 1 ORV shedding, or post-vaccination RV-IgA (Supplementary Fig. 2B and Supplementary Data 1–3).

**Geographic differences in microbiota development**. We sequenced 2137 separate faecal samples from the study population, of which 2086 yielded high-quality microbiota profiles (≥25,000 sequences after quality filtering; 143,123 ± 136,113 [mean ± s.d.] sequences per sample). Microbiota profiles were consistent across sequencing runs and facilities (explored by independently re-sequencing 10% of samples; Supplementary Fig. 3). Infant samples contained 461 genera, of which a small

number were dominant (Supplementary Fig. 4). The trajectory of microbiota development was highly distinct to each cohort, with significant deviations apparent as early as the first week of life. Microbiota diversity was significantly higher in Malawi than both other cohorts (Fig. 3A), although this discrepancy receded with increasing age. Inter-individual differences accounted for 58% of variation in microbiota composition based on permutational multivariate analysis (PERMANOVA; $p = 0.001$, 999 permutations), while country accounted for 6–9% of variation depending on age (Fig. 3B). Based on longitudinal mixed-effects models, 12 genera (including *Prevotella*, *Sutterella*, *Corynebacterium*, and *Acinetobacter*) were enriched across infancy in Malawi compared with both India and the UK (Supplementary Fig. 5). On the other hand, *Bifidobacterium*, *Enterococcus*, *Staphylococcus*, and *Streptococcus* were enriched in Indian infants compared with both other cohorts, while *Bacteroides*, *Citrobacter*, *Enterobacter*, *Haemophilus*, and *Klebsiella* were enriched in the UK (Fig. 3B and Supplementary Fig. 5). Numerous other discriminant taxa were identified during cross-sectional analyses of genus prevalence and abundance at each sampling timepoint (Supplementary Fig. 5C

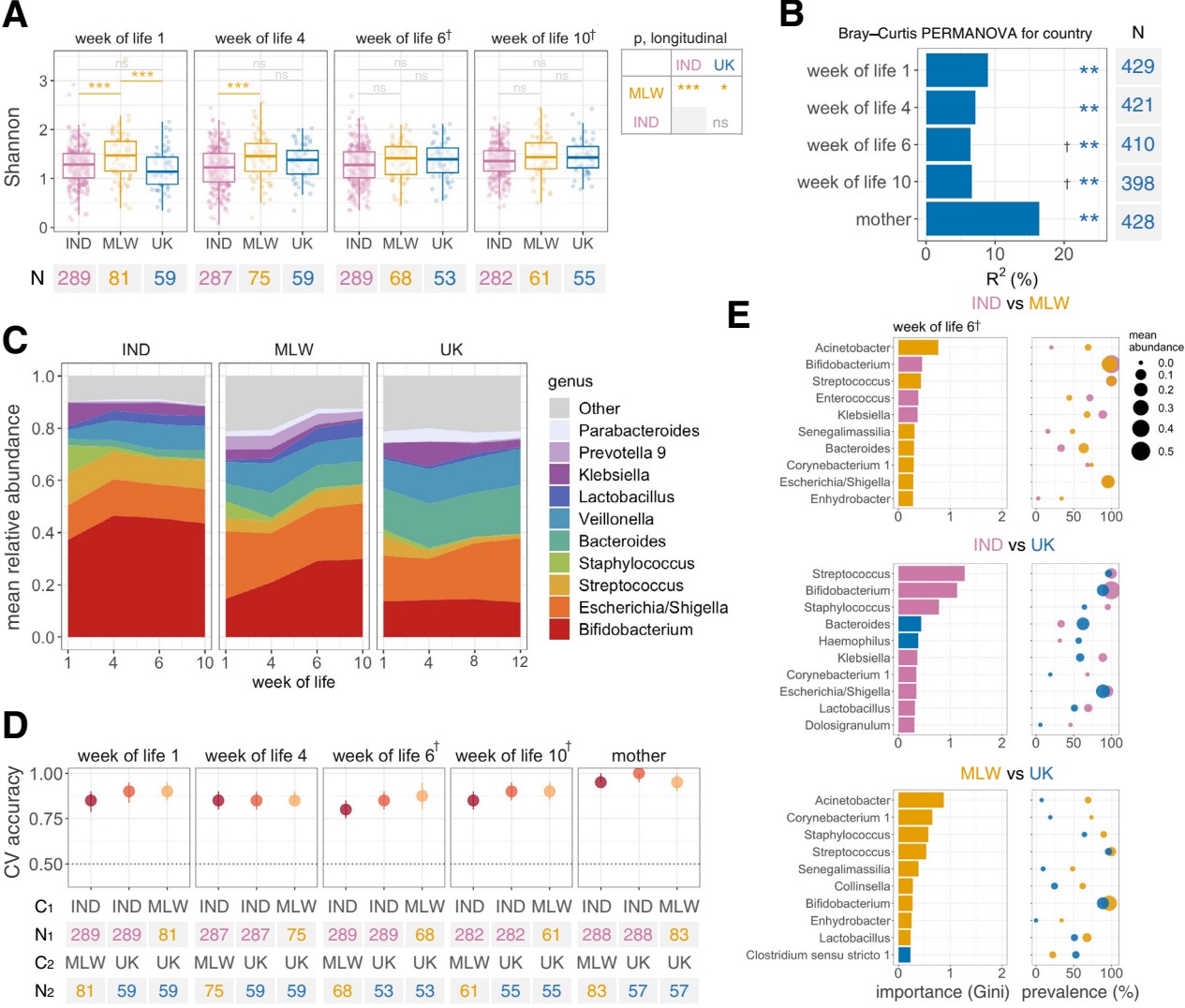

**Fig. 3 Geographic differences in microbiota development. A** Longitudinal analysis of alpha diversity. Shannon index was calculated at genus level. Cross-sectional comparisons were performed using ANOVA with post-hoc Tukey tests. Longitudinal comparisons were performed using mixed-effects regressions with week of life as a covariate and study ID as a random effect. Pairwise longitudinal comparisons between countries were FDR corrected. See Fig. 1 legend for box plot parameters. **B** Proportion of variation in microbiota composition associated with country. $R^2$ and statistical significance were determined by PERMANOVA using genus-level unweighted Bray–Curtis distances. **C** Longitudinal plot of mean genus abundances. Genera are displayed if they were present with a mean relative abundance of ≥5% in at least one country at one or more timepoints. **D** Cross-validation accuracy of Random Forests for prediction of country. Genus relative abundances served as input for each model. Median out-of-bag accuracy (proportion correctly assigned) and interquartile range across 20 iterations of 5-fold cross-validation are displayed. A random subset of 50 samples per country was used for each iteration. **E** The 10 most important genera selected by Random Forests for discriminating infants by country at the time of the first dose of ORV. Mean cross-validation importance scores based on Gini index are depicted alongside the prevalence and mean abundance of the corresponding genera in each country. C, country; CV, cross-validation; IND, India; MLW, Malawi; ns, not significant; †, +2 weeks in UK due to later vaccination schedule; *$p < 0.05$; **$p = 0.001$; ***$p < 0.0005$.

and Supplementary Data 4). Moreover, using the machine-learning method Random Forests, samples could be accurately distinguished by country at all timepoints based on genus relative abundances (median cross-validation accuracies of 80–90% for infant samples; baseline accuracy 50%; Fig. 3D, E).

To explore cofactors associated with the developing microbiota in each cohort, we performed an exploratory analysis of alpha and beta diversity among samples collected at the time of the first ORV dose (Supplementary Fig. 6A). Based on PERMANOVA of genus-level unweighted Bray–Curtis distances, microbiota composition in India was significantly correlated with α1AT ($R^2$ 3.6%), breastfeeding status ($R^2$ 2.0%), age at vaccine delivery ($R^2$ 1.3%), and delivery mode ($R^2$ 1.1%), though only breastfeeding

status was significantly correlated with microbiota diversity (mean [s.d., $n$] Shannon index of 1.2 [0.4, 248] in exclusively breastfed infants vs 1.5 [0.4, 41] in infants with partial or no breastfeeding; FDR $p < 0.001$). Neonatal rotavirus infection was not significantly correlated with microbiota composition or diversity (Supplementary Fig. 6B). In Malawi, microbiota composition was associated with maternal RV-IgA level ($R^2$ 4.8%), α1AT ($R^2$ 4.5%), HIV exposure ($R^2$ 4.5%), and maternal age ($R^2$ 4.1%). α1AT and maternal RV-IgA were positively correlated with microbiota diversity. No covariates were significantly associated with alpha or beta diversity in the UK after FDR correction, although fewer variables were measured in this cohort.

**Microbiota composition versus ORV response**. In both India and Malawi, microbiota diversity was negatively correlated with ORV seroconversion (Fig. 4A) and post-vaccination RV-IgA concentration (Supplementary Fig. 7A). In India, the discrepancies by seroconversion status were apparent in longitudinal models as well as cross-sectional comparisons at weeks of life 4 and 6. Interestingly, stratified analyses in India revealed the discrepancies to be specific to infants with no neonatal rotavirus infection (Fig. 4A and Supplementary Fig. 7A). These associations were robust in sensitivity analyses restricted to exclusively breastfed infants or those born by vaginal delivery (p values of 0.008 and 0.004 for respective longitudinal models of microbiota diversity; Supplementary Fig. 8). Similar discrepancies in microbiota diversity were evident in Malawian infants, albeit specifically at the time of the first dose of ORV. By contrast, microbiota diversity was not strongly associated with ORV seroconversion or post-vaccination RV-IgA concentration in the UK, and was not significantly associated with ORV shedding after dose 1 in any cohort (Supplementary Fig. 9A).

These results are corroborated by cross-sectional analyses of beta diversity. Based on PERMANOVA of unweighted Bray–Curtis distances, microbiota composition was significantly but modestly ($R^2 < 4\%$) correlated with seroconversion (Fig. 4B) and post-vaccination RV-IgA (Supplementary Fig. 7B) in Malawian infants as well as Indian infants without neonatal rotavirus infection. However, we did not observe clear differences in taxonomic composition according to ORV response. Cross-sectional Random Forests models based on genus or ribosomal sequence variant (RSV) abundances failed to accurately predict seroconversion (Fig. 4C), post-vaccination RV-IgA (Supplementary Fig. 7C), or ORV shedding after dose 1 (Supplementary Fig. 9C). After FDR correction, we did not observe significant differences in the prevalence (based on Fisher's exact test) or abundance (based on Aldex2) of individual genera or RSVs according to ORV outcome. Likewise, longitudinal models of common genera (present in ≥20% of samples) revealed numerous age-associated changes in abundance, but very few taxa with differential abundance according to ORV outcome (Supplementary Table 2). Overall, while microbiota diversity appears to be significantly higher among infants in India and Malawi who fail to respond to ORV, we did not observe consistent discrepancies in specific bacterial taxa according to vaccine outcome.

**Multivariate analysis**. To delineate the contribution of different risk factors in shaping ORV response, we used the machine-learning algorithm Random Forests to predict ORV outcome based on a combination of: (i) demographic and baseline health variables ($n = 18$; see Supplementary Data 2 for complete list); (ii) exposure/antibody data ($n = 12$), including EED markers and maternal antibody concentrations; and (iii) genus relative abundances at the time of the first ORV dose (week of life 6; $n = 55$). These analyses focused on Indian infants given the larger size of this cohort ($n = 249$ with complete data). Moreover, we included analyses stratified by neonatal rotavirus infection status given its modifying effect on the associations reported above. Models exhibited poor accuracy for the prediction of seroconversion and ORV shedding after dose 1 (Fig. 5A). Post-vaccination RV-IgA concentration was predicted with modest accuracy ($R^2$ of 37.2%; interquartile range [IQR] 31.4–42.5%)—an effect driven primarily by neonatal rotavirus infection (Fig. 5B). Prediction of post-vaccination RV-IgA was markedly reduced in the stratified analyses ($R^2$ of 16.1% [9.6–22.1%] and 3.7% [0.3–5.1%] in infants with and without neonatal infection, respectively). Maternal RV-IgG was the most important predictor of post-vaccination RV-IgA among infants with neonatal infection, whereas multiple

maternal antibody and microbiota variables were among the top-ranking features for infants without neonatal infection (Fig. 5B).

We complemented the machine-learning models with multi-variate regressions. Covariates with a $p$ value of <0.05 in univariate analyses were selected for inclusion in these models. Consistent with the Random Forests outputs, few significant predictors were identified for multivariate models of seroconversion among Indian infants (Supplementary Data 2). ORV shedding after dose 1 in this cohort was negatively correlated with neonatal rotavirus infection (RR 0.51 [0.30–0.80]) and maternal RV-IgA (RR 0.68 [0.53–0.85]), and was higher in the sequentially recruited cohort of IPV recipients (RR 2.18 [1.41–3.08]; see Supplementary Data 1 for full results). Neonatal rotavirus infection was the most important predictor of post-vaccination RV-IgA in Indian infants, while breastfeeding practice, linear growth, maternal RV-IgG, and polio vaccine schedule were also significant predictors for this outcome (Table 2; see Supplementary Data 3 for full univariate and multivariate results). Associations varied according to neonatal rotavirus infection status—whereas maternal RV-IgG concentration and breastfeeding status were the strongest predictor of post-vaccination RV-IgA among infants with neonatal rotavirus infection, significant predictors of RV-IgA in those lacking neonatal infection included microbiota diversity, house type, and access to treated water. Overall, the multivariate regressions offered a predictive accuracy commensurate with or exceeding that of Random Forests ($R^2$ of 43% in the complete cohort and 11–19% in the stratified analyses for RV-IgA regressions; Table 2).

## Discussion

ORV is unique among oral vaccines in being routinely administered in both high-income countries and LMICs. As such, the vaccine provides a valuable tool for probing the divergent trajectories of the developing immune system in different settings. Here, we report on the extent to which several key determinants of infant immune development differ among three disparate populations, and the degree to which these factors shape ORV response.

As expected, Rotarix response was significantly impaired among infants in Malawi and India. While the later vaccination schedule in the UK (8/12 weeks vs 6/10 weeks) may contribute to these differences by providing more time for passively acquired maternal antibodies to wane, overall shedding rates after both doses in Malawi and India still fell short of the near-ubiquitous dose 1 shedding observed in the UK, highlighting the barriers to ORV that emerge within the first weeks of life in LMICs. Seroconversion rates in India and Malawi were broadly consistent with those previously reported in LMICs[15,16], but lower than expected in the UK[17]. These findings highlight the suboptimal nature of serum RV-IgA as a correlate of protection. While higher RV-IgA concentrations correlate with a lower risk of subsequent rotavirus infection, these associations often fail to hold when considering protection against rotavirus-associated gastroenteritis[18]. Although ORV shedding has not been validated as a correlate of protection, our study highlights the potential value of this outcome in capturing divergent ORV responses within and between populations. We advocate the further use of vaccine virus shedding in the week after administration (at multiple timepoints where possible) as an adjunctive measure of ORV response in future trials. Faecal RV-IgA has also been shown to correlate with protection against rotavirus illness and merits further consideration in infant vaccine trials[19].

For a risk factor to account for broad geographic trends in ORV efficacy, it is reasonable to expect it to be more common or observed at higher levels in LMICs than high-income countries.

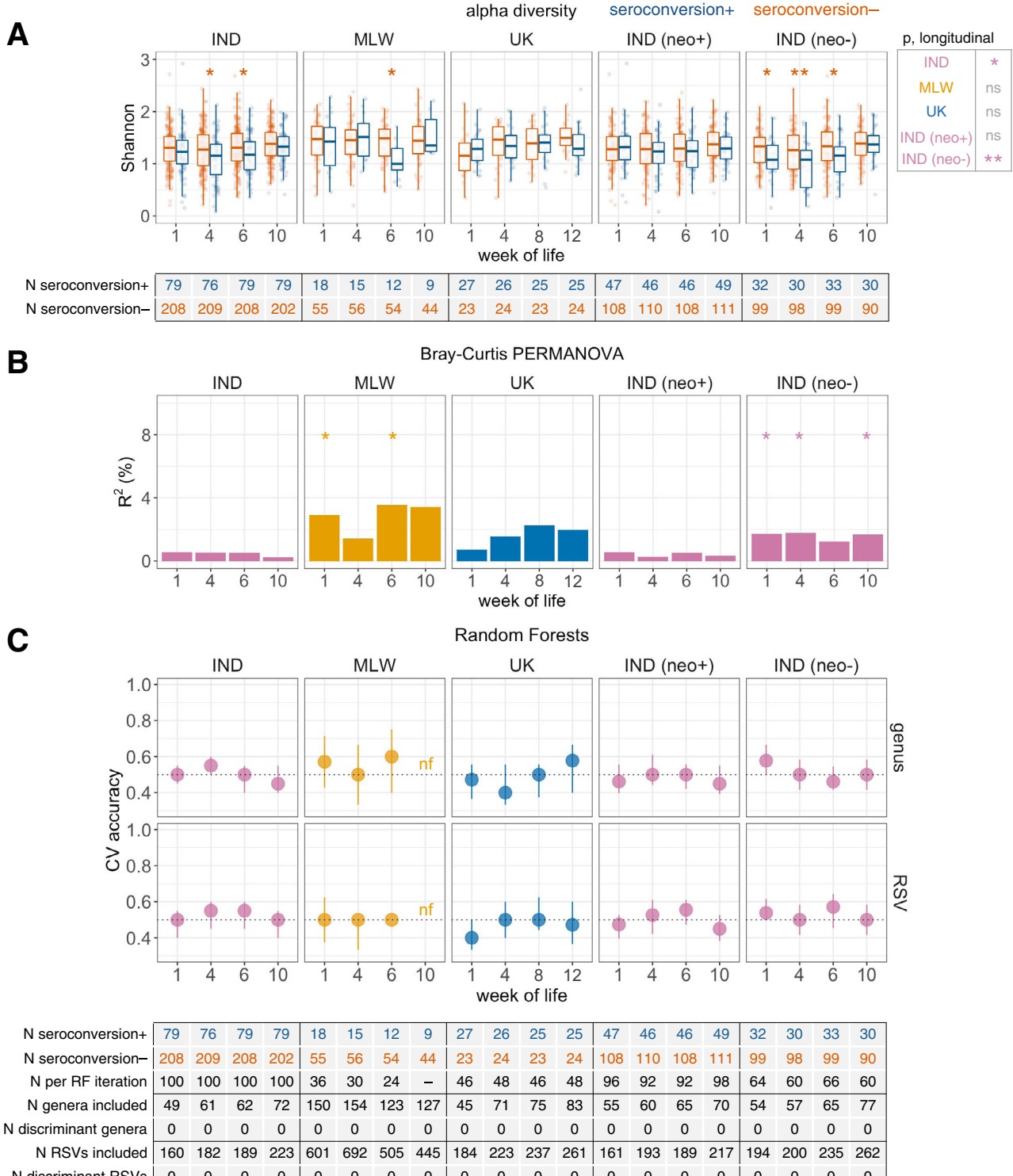

**Fig. 4 Association between microbiota development and oral rotavirus vaccine seroconversion. A** Longitudinal analysis of alpha diversity. Shannon index was calculated at genus level. Cross-sectional comparisons were performed using logistic regression. Longitudinal comparisons were performed using mixed-effects models including week of life as a covariate and study ID as a random effect. See Fig. 1 legend for box plot parameters. **B** Proportion of variation in microbiota composition associated with seroconversion. $R^2$ and statistical significance were determined by PERMANOVA using genus-level unweighted Bray-Curtis distances. **C** Cross-validation accuracy of Random Forests for prediction of seroconversion. Models were fitted at genus and RSV level. Median out-of-bag accuracy (proportion correctly assigned) and interquartile range across 20 iterations of 5-fold cross-validation are displayed. Each iteration included an equal number of responders and non-responders (50 per group where possible, or else the number in the minority group if this was <50). For each cross-sectional comparison, taxa were classified as discriminant if they had an FDR-adjusted $p$ value of <0.05 based on either two-sided Fisher's exact test (differences in prevalence) or Aldex2 with two-sided Wilcoxon rank-sum test (differences in abundance). CV, cross-validation; IND, India; MLW, Malawi; neo+, infected with rotavirus neonatally (defined by detection of rotavirus shedding in week of life 1 or baseline seropositivity); neo−, uninfected with rotavirus neonatally; nf, not fitted as <10 responders; ns, not significant; RF, Random Forests; RSV, ribosomal sequence variant; *$p$ < 0.05; **$p$ < 0.005; ***$p$ < 0.0005.

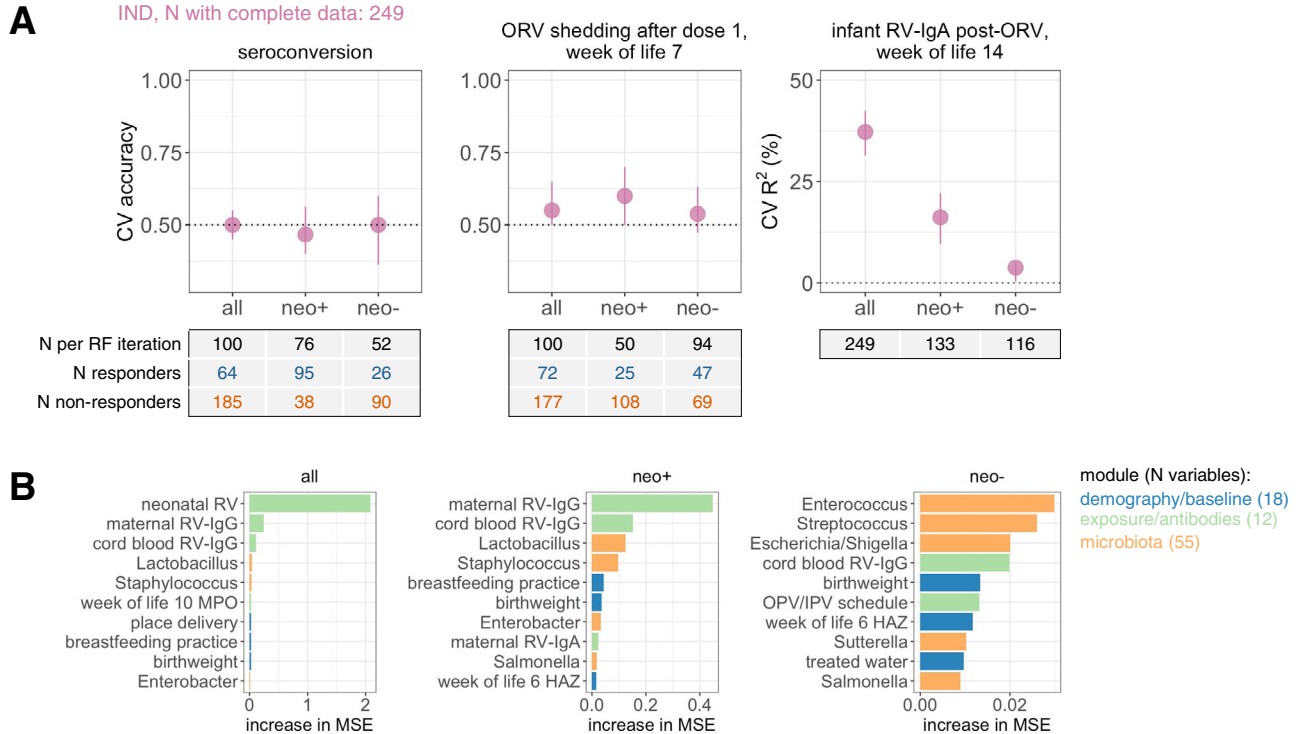

**Fig. 5 Integrated analysis for prediction of oral rotavirus vaccine response in indian infants. A** Cross-validation accuracy of Random Forests models for prediction of ORV response based on demographic variables, exposure/antibody variables, and genus relative abundances at the time of dose 1 (week of life 6). Infants with complete data were included. Median out-of-bag accuracy and interquartile range across 20 iterations of 5-fold cross-validation are displayed. For binary outcomes (seroconversion and shedding), each iteration included an equal number of responders and non-responders (50 per group where possible, or else the number in the minority group if this was <50). Out-of-bag accuracy reflects the proportion of infants correctly assigned (binary outcomes) or $R^2$ for predicted vs observed outcome (RV-IgA). **B** The top 10 variables selected by Random Forests for prediction of post-vaccination RV-IgA concentration based on mean cross-validation importance score. CV, cross-validation; HAZ, height-for-age $Z$ score; IND, India; IPV, inactivated poliovirus vaccine; neo+, infected with rotavirus neonatally (defined by detection of rotavirus shedding in week of life 1 or baseline seropositivity); neo−, uninfected with rotavirus neonatally; MPO, myeloperoxidase; MSE, mean squared error; OPV, oral poliovirus vaccine; ORV, oral rotavirus vaccine; RF, Random Forests; RV, rotavirus.

This was not the case for several of the factors examined in this study. Maternal serum RV-IgA levels were comparable in the UK and India (although significantly higher in Malawi), while breastmilk RV-IgA levels were higher in the UK than in India. These results are somewhat surprising given that recent rotavirus exposure and consequent boosting of antibody levels might be expected to occur more frequently among mothers in LMICs with a high rotavirus disease burden. In previous studies, RV-IgA concentrations were significantly higher in India than the USA, with intermediate levels in South Africa[20,21]. Our findings caution against the broader extrapolation of these trends. The high maternal RV-IgA levels in the UK may reflect the immunogenic nature of early rotavirus exposures and the subsequent boosting of antibodies by mild or asymptomatic re-infection[22]. Nutritional stress has also been linked with lower secretory IgA levels in breastmilk[23,24]. Consistent with this, we observed maternal breastmilk/serum RV-IgA ratios to be lowest in India and highest in the UK, with intermediate levels in Malawi.

Although maternal antibody concentrations did not follow a clear geographic gradient, our findings confirm the potential inhibitory effect of these antibodies on ORV among infants in LMICs. Whereas previous studies of this phenomenon have typically focused on immunogenicity endpoints[8,25], we observed a link with both shedding and immunogenicity, suggesting that the effect of maternal antibodies may be mediated in part through a reduction in ORV shedding. This could potentially involve the transudation of transplacental RV-IgG across the mucosal

epithelium or the direct neutralisation of vaccine viruses by RV-IgA in breastmilk. Our findings are also consistent with the inhibition of downstream antigen processing by transplacental antibodies, as evidenced by the strong inverse correlation between maternal RV-IgG and infant RV-IgA formation pre-vaccination in Indian infants with neonatal rotavirus infection. Overall, one could reasonably envision a combination of upstream inhibition of ORV shedding (via RV-IgA or transudated RV-IgG) and downstream inhibition of antigen processing (via circulating RV-IgG) that synergistically inhibit ORV outcome. However, these mechanisms alone are clearly insufficient to prevent a robust ORV response given that the vaccine was ubiquitously shed and was most immunogenic in the UK, where maternal antibody levels were high and >75% of infants were breastfed. Moreover, exclusive breastfeeding was positively correlated with post-vaccination RV-IgA in Indian infants, suggesting that any inhibitory effect of RV-IgA in milk are offset by the potential benefits of breastfeeding for ORV, such as the buffering of gastric acid.

We considered several markers as a proxy for EED. Although α1AT concentrations were significantly lower in the UK than both other cohorts, MPO levels were similar in the UK and Malawi, calling into question the extent to which this marker accurately captures the onset of EED. We did not observe a significant relationship between any of the measured EED markers and ORV outcome, in keeping with the growing literature in this field[11,26]. Together, these findings imply either that EED is not a primary driver of oral vaccine failure and/or that the condition is

**Table 2 Multivariate regression for post-vaccination rotavirus-specific IgA concentration in Indian infants.**

| variable | univariate | | multivariate | |
|---|---|---|---|---|
| | beta (95% CI) | p | beta (95% CI) | p |
| *IND* | | | *n* = 286; adjusted $R^2$: 43.4% | |
| Delivery mode (caesarean vs vaginal) | 2.04 (1.19–3.49) | 0.010 | 1.45 (0.93–2.28) | 0.102 |
| Delivery place (tertiary vs non-tertiary) | 2.69 (1.72–4.22) | <0.001 | 1.08 (0.72–1.64) | 0.701 |
| Breastfeeding practice (exclusive vs non-exclusive) | 2.29 (1.18–4.45) | 0.015 | 1.89 (1.11–3.22) | 0.019 |
| Height-for-age Z score (week of life 6) | 1.32 (1.07–1.63) | 0.008 | 1.31 (1.10–1.55) | 0.002 |
| Exposed to rotavirus pre-vaccination | 9.34 (6.37–13.71) | <0.001 | 8.48 (5.67–12.67) | <0.001 |
| Polio vaccine schedule: bOPV only | ref | — | ref | — |
| bOPV/tOPV mixed | 0.54 (0.28–1.05) | 0.070 | 0.48 (0.28–0.85) | 0.011 |
| tOPV only | 0.49 (0.25–0.96) | 0.038 | 0.43 (0.25–0.75) | 0.003 |
| IPV | 0.55 (0.31–0.97) | 0.041 | 0.77 (0.49–1.22) | 0.265 |
| House made of permanent materials (yes vs no) | 2.03 (1.26–3.26) | 0.004 | 1.28 (0.86–1.91) | 0.228 |
| Access to treated water (yes vs no) | 1.66 (1.04–2.67) | 0.035 | 1.35 (0.93–1.96) | 0.120 |
| Maternal RV-IgG (IU/ml), log-transformed | 0.59 (0.47–0.73) | <0.001 | 0.59 (0.49–0.71) | <0.001 |
| *IND neo+* | | | *n* = 166; adjusted $R^2$: 19.4% | |
| Breastfeeding practice (exclusive vs non-exclusive) | 4.08 (1.81–9.18) | 0.001 | 3.45 (1.62–7.33) | 0.001 |
| Height-for-age Z score (week of life 6) | 1.39 (1.08–1.79) | 0.012 | 1.25 (0.99–1.58) | 0.061 |
| Maternal RV-IgG (IU/ml), log-transformed | 0.50 (0.38–0.65) | <0.001 | 0.54 (0.41–0.70) | <0.001 |
| *IND neo–* | | | *n* = 124; adjusted $R^2$: 10.8% | |
| Delivery mode (caesarean vs vaginal) | 2.58 (1.31–5.06) | 0.006 | 1.74 (0.81–3.73) | 0.153 |
| Delivery place (tertiary vs non-tertiary) | 1.93 (1.04–3.58) | 0.036 | 1.17 (0.60–2.28) | 0.637 |
| House made of permanent materials (yes vs no) | 1.84 (1.11–3.07) | 0.019 | 1.72 (1.02–2.92) | 0.043 |
| Access to treated water (yes vs no) | 2.04 (1.22–3.42) | 0.007 | 1.82 (1.08–3.08) | 0.026 |
| Shannon (week of life 6) | 0.52 (0.28–0.99) | 0.047 | 0.48 (0.26–0.90) | 0.021 |

Log-transformed RV-IgA concentration measured 4 weeks after dose 2 (week of life 14) served as the dependent variable. Variables with a *p* value of <0.05 during univariate analyses are shown here. Full univariate regression results are provided in Supplementary Data 3. Infants with complete data for all selected covariates were included in multivariate models. Equivalent results for dose 1 ORV shedding and seroconversion are provided in Supplementary Data 1 and 2, respectively.

not accurately captured by the current suite of faecal and systemic markers. However, we observed a negative correlation between post-vaccination RV-IgA and height-for-age Z score in Indian infants, corroborating recent data from Zimbabwe linking growth deficits with impaired ORV response[14].

A key finding of translational relevance is the highly immunogenic nature of neonatal rotavirus infection in India. Neonatal infection was detected in almost half of the infants in this cohort, and these infants exhibited significantly higher final RV-IgA concentrations than uninfected infants. Machine-learning models corroborated this by selecting neonatal rotavirus infection status as the most important determinant of final RV-IgA concentration out of 85 input variables. Neonatal infection with the G10P[11] strain was not sensitive to the inhibitory effect of maternal antibody levels (a trait also reported for the G3P[6] oral human neonatal rotavirus vaccine candidate RV3-BB[27]), potentially reflecting serotype-specific adaptation to the newborn gut[28] alongside reduced exposure to RV-IgA and other antiviral compounds if infection occurs prior to breastfeeding. Trials exploring the shedding and immunogenicity of Rotarix (a G1P[8] strain) when administered neonatally would help distinguish these possibilities. In Indonesia, RV3-BB efficacy against severe rotavirus gastroenteritis was 75% when administered according to a neonatal schedule (weeks 1, 8, and 14) and 51% when administered on a later schedule (weeks 8, 14, and 18), highlighting the efficacy of schedules that include ORV at birth[29].

We observed marked geographic discrepancies in composition of the bacterial microbiota. Microbiota diversity was significantly higher among Malawian infants, while Indian infants were distinguishable by their high *Bifidobacterium* abundance. The latter is interesting given that *Bifidobacterium* has previously been suggested to enhance immunological memory responses to a range of vaccine targets[30], but was least abundant among infants

in the UK that were most responsive to ORV. Similarly, we failed to recapitulate other associations that have been documented between bacterial microbiota composition and ORV outcome, such as the higher abundance of Bacteroidetes among non-responders in Ghana[12]. Although we did not record any clear discrepancies in taxon abundances according to ORV response, we observed a significant negative correlation between microbiota diversity and vaccine immunogenicity among Indian and Malawian infants. In India, the correlation was strongest in infants lacking neonatal rotavirus infection, and was robust in sensitivity analyses restricted to exclusively breastfed or vaginally delivered infants. Given that infant microbiota diversity was comparable in India and the UK, this factor alone is clearly insufficient to account for broad geographic trends in ORV response. However, the observation of a consistent microbiota signature of impaired vaccine response across multiple LMICs is noteworthy. It is plausible that early-life exposure to greater microbial diversity (including members of the virome and eukaryome not measured here) may foster a state of hyporesponsiveness at the mucosal epithelium that impairs oral vaccine outcome in LMICs. Impaired vaccine response could thus be viewed as a counterpart to hyperresponsive immune states associated with low early-life microbiota diversity, such as atopic disease[31]. Our findings in Indian infants also support the notion that the neonatal period represents a key window of microbiome development[32]. Indeed, associations between microbiota diversity and ORV response in this cohort were apparent from the first week of life onwards. While a growing body of literature has emphasised the importance of microbiota composition in relation to health outcomes such as malnutrition among children in LMICs[33,34], studies focusing on the first days and weeks of life are presently lacking, and represent a crucial avenue of future research.

Despite the multiple demographic, immunological, and microbiota variables considered, our overall ability to account for discrepancies in ORV response within and between cohorts was limited. Alternative risk factors for impaired ORV outcome must therefore be considered. In particular, we would advocate further efforts to profile early-life host–microbe interactions in LMICs, including high-throughput measurements of breastmilk (e.g. metabolome, microbiome, and antibody functionality) and faecal samples (e.g. virome and metabolome). This will help to determine the extent to which metabolic and microbial exposures combine to inhibit ORV response over the early weeks of life. The epigenome merits further consideration given that BCG-induced epigenetic remodelling of monocytes has been linked with protection against heterologous viruses[35]. It is also possible that we considered the correct mechanisms but failed to capture them with sufficient granularity using the methods and samples available. Alternative EED markers may be required to accurately characterise the onset of this condition in early life. Moreover, given the potential for overall microbial load to vary substantially from infant to infant, absolute as opposed to relative quantitation of microbial abundances may be necessary to capture the interplay between the developing bacterial microbiota and ORV outcome[36].

Our study has several limitations. First, as noted above, RV-IgA is a suboptimal correlate of protection. We compensated for this by considering multiple endpoints, including seroconversion and post-vaccination shedding, though longer-term follow-up for rotavirus-associated gastroenteritis was beyond the scope of the study. Second, challenges in sample recruitment in Malawi meant that the final study population ($n = 119$) fell short of the target sample size ($n = 150$). This was exacerbated by incomplete sample availability for recruited infants, such that several comparisons in this cohort may have been underpowered. Relatedly, comparisons of post-vaccination RV-IgA across cohorts may have been underpowered given the lower-than-expected seroconversion rates in the UK. Finally, whilst the potential contribution of batch effects to the observed differences between populations cannot be completely ruled out, we went to considerable lengths to ensure reproducibility and comparability across sites, using the same reagent lots and standards. In the case of microbiota composition, we independently re-sequenced 10% of samples at a separate facility and recapitulated the original microbiota composition with high accuracy.

Notwithstanding these caveats, this study advances our understanding of the potential mechanisms influencing ORV response in several ways. By considering how several risk factors of ORV failure differ within and between populations, we have been able to place these mechanisms within a broader global context. Our study confirms the inhibitory effect of maternal antibodies on ORV immunogenicity in LMICs, and suggests this may be mediated partly by a reduction in ORV shedding. We observed that asymptomatic neonatal rotavirus infection was strongly associated with RV-IgA formation, providing further support for the potential of neonatal vaccination as a pragmatic approach to achieve greater rotavirus vaccine impact. Finally, while specific bacterial taxa do not appear to drive within-population differences in ORV response, high microbiota diversity in early life may contribute to the impaired efficacy of this vaccine in LMICs.

## Methods

**Study design**. Pregnant women were enrolled during the third trimester at three study sites: Blantyre (Malawi), Vellore (India), and Liverpool (UK). Women were included if they provided informed written consent to participate in the study and were willing to stay in the study area for 4 months following delivery. Exclusion criteria included: congenital immune deficiency; chronic renal or liver failure; other

chronic illnesses which may affect immune function; non-singleton pregnancy; low birthweight or pre-term birth (<34 weeks gestation); congenital anomalies and other neonatal complications requiring prolonged hospitalisation; and delivery by elective caesarean (UK only). Mothers provided a venous blood sample as well as a cord blood sample during birth and a stool sample during the week after delivery. Infants provided two blood samples (weeks of life 6 and 14 in India and Malawi; weeks of life 8 and 16 in the UK) and six stool samples over the course of the study (Fig. 1A), including samples at the time of each ORV dose (weeks of life 6 and 10 in Malawi and India; weeks of life 8 and 12 in the UK). For post-vaccination rotavirus shedding assays (weeks of life 7 and 11 in Malawi and India; weeks of life 9 and 13 in the UK), stool samples provided within ±1 days of the target were considered eligible for inclusion. For all other timepoints, serum and stool samples provided within ±3 days were considered eligible. Overall, we enrolled 664 mother–infant dyads (395 in India, 187 in Malawi, and 82 in the UK). The primary endpoint (measurement of seroconversion or dose 1 shedding) was reached by 484 dyads (307 in India, 119 in Malawi, and 60 in the UK). The study was approved by the Institutional Review Board at the Christian Medical College (CMC) in Vellore, the College of Medicine Research and Ethics Committee in Blantyre, and the North West—Liverpool East Research Ethics Committee in Liverpool. The trial is registered with the Clinical Trials Registry of India (CTRI/2015/11/006354) and the study protocol has previously been published[37].

**Sample processing and storage**. Whole blood was collected in anti-coagulation EDTA-tubes (BD) and stored at 4 °C for up to 12 h until collection by laboratory staff. Fractions were separated by centrifugation at 1500–2000 × $g$ in a benchtop centrifuge (Eppendorf). A disposable plastic transfer pipette was used to aspirate the plasma down to ~1 mm from the red blood cells. The plasma was aliquoted into two screw cap cryo-tubes (Starlab). The buffy coat and red blood cells were subsequently collected into separate cryovials and all samples were stored at −70 °C.

Stool and breastmilk samples were collected in sterile sample pots by participants and were shipped to the respective laboratory by courier within 24 h in the UK and within 4 h in India and Malawi (to account for ambient temperatures). Samples were kept at 4 °C for a maximum of 8 h until processing. Breastmilk samples were stored in 2 ml aliquots in SuperLock tubes (Starlab) at −70 °C for maternal RV-IgA analysis. Two 10% stool suspensions were prepared in sterile phosphate-buffered saline (PBS) for assessing rotavirus shedding. Further aliquots of neat stool were stored in 2 ml SuperLock tubes (Starlab) at −70 °C for microbiota analysis and inflammatory biomarker measurement. Stool samples were stored at −70 °C for a maximum of 2 weeks prior to DNA extraction for microbiota analysis.

**Rotavirus-specific antibodies**. RV-IgA and RV-IgG laboratory assays were conducted at CMC, Vellore, for samples from all three study sites using a custom antibody-sandwich ELISA[38]. However, a technical issue with the assay precluded the inclusion of RV-IgG data from the UK and Malawi. Briefly, 96-well plates (Costar) coated with rabbit hyperimmune serum to rotavirus (at a dilution of 1 in 1500 for RV-IgA and 1 in 1000 for RV-IgG) were incubated with purified cell culture lysates (WC3, at a dilution of 1 in 5 for RV-IgA and 1 in 4 for RV-IgG) or mock-infected MA104 cells (at a dilution of 1 in 5 for RV-IgA and 1 in 4 for RV-IgG). Serial dilutions of standard and test sera were added followed by biotinylated rabbit anti-human IgA (Jackson ImmunoResearch Laboratories, at a dilution of 1 in 3000) for detection of RV-IgA and biotinylated rabbit anti-human IgG (Vector Laboratories, at a dilution of 1 in 800) for detection of RV-IgG. Absorbance was subsequently read at 492 nm. Background-corrected optical density values from sample wells were compared with the standard curve and IgA or IgG concentration was determined based on derived units of IgA or IgG arbitrarily assigned to the respective standard curve, with a minimum detection limit of 1 IU/ml. Seroconversion was defined as detection of RV-IgA at ≥20 IU/ml in previously seronegative infants or a 4-fold increase in RV-IgA concentration among infants who were seropositive at baseline.

**Rotavirus shedding**. Quantification of rotavirus shedding was performed at each study site. RNA was extracted using the RNeasy mini kit (Qiagen) following the manufacturer's instructions. Briefly, 10% PBS stool suspensions were defrosted on ice, vortexed, and briefly centrifuged (10,000 × $g$ for 10 s) to pellet larger debris. 250 μl of β-ME in buffer RTL was added to 250 μl of supernatant. The mixture was then vortexed before the addition of 250 μl of 70% molecular-grade ethanol. The RNeasy Mini Handbook (5th Edition, October 2013) was followed for subsequent steps, and RNA was eluted in 50 μl RNase-free water. cDNA was generated immediately after RNA extraction. The master mix for cDNA conversion comprised: 7 μl of 10× Taq DNA PCR buffer (Invitrogen), 7 μl of 50 mM MgCl$_2$ (BIOLINE), 1 μl of Random Primers (Invitrogen), 2 μl of 10 mM dNTP (Invitrogen), 1 μl of DTT (Invitrogen), 2 μl of SuperScript III reverse transcriptase (Invitrogen), and 10 μl of RNase-free water (Invitrogen). 40 μl of eluted RNA was denatured at 95 °C for 5 min and subsequently cooled on ice for 2 min. 30 μl of master mix was added to each RNA sample, mixed gently by pipetting, and centrifuged briefly. The tube was placed in a thermal cycler and incubated at 25 °C for 10 min, 37 °C for 1 h, then 95 °C for 5 min. The cDNA was stored at −20 °C.

Previously validated primers and probes were used for the qPCR-based detection of rotavirus NSP2[39] and VP6[40] in two separate reactions, applying

maximum Ct thresholds of 40 and 35, respectively. cDNA and reagents were thawed on ice. The master mix for qPCR comprised: 12.5 µl of Platinum qPCR supermix-UDG 2x (Invitrogen), 2 µl of 10 µM forward primer (5′-GAACTTCCT TGAA-TATAAGATCACACTGA-3′ [NSP2] or 5′-GACGGVGCRACTACATGG T-3′ [VP6]), 2 µl of 10 µM reverse primer (5′-TTGAAGACGTAAATGCATACC AATTC-3′ [NSP2] or 5′-GTCCAATTCAT-NCCTGGTG-3′ [VP6]), 0.25 µl of 10 µM probe (FAM5′-TCCAATAGATTGAAGTCAGTAACGT-TTCCA-3′MGB [NSP2] or FAM5′-CCACCRAAYATGACRCCAGCNGTA-3′MGB [VP6]), and 6.25 µl of nuclease-free water (ThermoFisher Scientific). In a 96-well plate, 23 µl of master mix was combined with 2 µl of cDNA. Six standards for the target gene as well as a no-template control and a positive control were added to each PCR plate. The plate was sealed with a Microseal 'C' Film (Bio-Rad) and subjected to the following cycles: initial UDG-incubation 50 °C for 2 min; denaturation for 95 °C for 2 min; then 40 cycles of 95 °C for 15 s and 60 °C for 1 min. Data were considered valid if amplification was absent in the negative control and present in the positive control. qPCR standards consisting of TOPO-TA plasmid constructs containing either NPS2 or VP6 gene amplicons (genotype 1) were produced at the University of Liverpool and distributed to the other study sites.

**Inflammatory biomarkers.** For infant stools collected before each dose of ORV, MPO and α1AT concentrations were measured by ELISA at each site. ELISA kits with the same lot number were distributed across the study sites. Stool samples were defrosted on ice and 100 mg then aliquoted into a polypropylene tube. ELISAs were carried out following the manufacturer's instructions (IDK 2018 instructions for MPO and BioVendor RIC6200 for α1AT). Paired samples from a given infant were run on the same plate, and MPO and α1AT assays were carried out in tandem to avoid freeze—thaw cycles. For week 6 serum samples of Indian and Malawian infants, α1AG was measured at CMC, Vellore, using a commercial ELISA kit (abcam) according to the manufacturer's instructions. α1AG assays were not performed for the UK so that the limited sample volumes for this population could be prioritised for RV-IgA assays.

**DNA extraction from stool.** DNA extraction from stool was carried out using the QIAamp DNA stool mini kit (Qiagen) according to the manufacturer's instructions (QIAamp DNA Stool Handbook 06/2012), with several modifications to increase DNA yield. Briefly, the vortexed suspension of ASL buffer and stool was added to 370 mg of 0.1 mm zirconia-silicate beads (BioSpec) with 1.67 µl of lyzozyme at 30 mg/ml (Sigma-Aldrich). Samples were then incubated at 37 °C on a shaker (Eppendorf Thermo Mixer Comfort) at 250–300 rpm for 10 min. 10 µl of protei-nase K (Qiagen), 50 µl of 10% sodium dodecyl sulphate (Sigma-Aldrich), and 20 µl of RNase A (1 mg/ml; Sigma-Aldrich) were added before incubation on a heating block (Eppendorf Thermo Mixer Comfort) at 70 °C for 10 min. After allowing the samples to cool for 3 min, bead beating was performed for 5 min using a Tissue Lyser II (Qiagen) at 25 Hz. Samples were microcentrifuged at $16,100 \times g$ for 1 min and the supernatant was then transferred into a new microcentrifuge tube con-taining an InhibitEX tablet and vortexed for 1 min. The suspension was incubated at room temperature for 1 min, centrifuged at $16,100 \times g$ for 5 min, and the supernatant (~600 µl) then transferred into a new 2 ml tube. The supernatant was centrifuged at $16,100 \times g$ for 3 min, then 400 µl added to a 2 ml tube containing 400 µl of buffer AL. 400 µl of ethanol (96–100%) was added to the lysate and mixed by vortexing. The manufacturer's instructions were then resumed until elution in 50 µl of AE Buffer. An extraction control was carried out with each batch of DNA extraction at all sites. Extracted DNA was shipped from India and Malawi to the University of Liverpool on dry ice for 16S rRNA sequencing.

**Amplicon generation for 16S rRNA sequencing.** Amplicons spanning 16S rRNA gene variable regions 3 and 4 (primers 309F 5′-ACTCCTACGGGAGGCAGCAG-3′ and 819 R 5′-GGACTACHVGGGTWTCTAAT-3′) were produced following the established Illumina protocol (Illumina, 16S Metagenomic Sequencing Library Preparation Protocol Part # 15044223 Rev. B) with the following amendments: 1 µl of DNA was used as a starting template; NEBNext Q5 Hot Start HiFi PCR Master Mix (NEB) was used for amplicon and indexing PCRs; PCR reaction volumes were halved to 25 µl; primers were used at a concentration of 10 µM; and 20 µg of molecular-grade bovine serum albumin (NEB) was added to the amplicon PCR master mix to mitigate the effect of PCR inhibitors. Cycling conditions for amplicon PCR involved: denaturation at 98 °C for 30 s; 10 cycles of 98 °C for 10 s, 55 °C for 15 s, and 72 °C for 40 s; and final extension at 72 °C for 60 s. Cycling conditions for indexing PCR involved: denaturation at 98 °C for 3 min; 15 cycles of 98 °C for 10 s, 55 °C for 15 s, and 72 °C for 40 s; and final extension at 72 °C for 5 min. AMPure XP beads were substituted with a custom preparation of Sera-Mag SpeedBeads Protein A/G particles (MERCK/GE Healthcare) throughout the protocol. To determine volumes required for equimolar pooling, amplicon concentrations were determined by Quanti-it (ThermoFisher Scientific) and size distributions by fragment analysis on a 5300 Fragment Analyzer System (Agilent). Amplicons were pooled using a mosquito X1 (TTP Labtech) liquid handling robot. The final library underwent size selection to remove potentially contaminating primer-dimers and genomic DNA using the Pippin Prep (Sage Science) 1.5% Agarose Gel Cassette (Labtech).

Overall, we sequenced amplicons from 2138 samples across 14 Illumina HiSeq2500 lanes (v2 chemistry, 600 cycles in rapid run mode) and one Illumina

MiSeq lane (v3 chemistry, 600 cycles). Since enrolment and sample collection across the three sites were not synchronised, sequencing was batched by geographic origin according to sample availability. Samples from a mother–infant pair were processed on the same PCR plate. Each PCR plate contained a no-template PCR control, stool controls provided by a mother and infant in the UK who were not enrolled in the study, DNA from a mock community (Zymo Research D6306), and a pool of extraction controls corresponding to the samples contained on the PCR plate. Final libraries consisted of up to four PCR plates (384 amplicons). Eight samples were excluded from the analysis owing to the presence of significant amplification from their corresponding extraction controls.

**Illumina sequencing of 16S rRNA libraries.** The quantity and quality of each amplicon pool was assessed by Qubit dsDNA High Sensitivity Assay Kit (Ther-moFisher Scientific), Bioanalyzer High Sensitivity DNA Kit (Agilent), and qPCR using the KAPA Library Quantification Kit (Roche) on a Light Cycler LC480II (Roche) according to the manufacturer's instructions. qPCR data were used to calculate sample molarity. 5 µl of the final pool was denatured for 5 min at room temperature using 5 µl of freshly diluted 0.1 N sodium hydroxide (Illumina). The reaction was subsequently terminated by the addition of HT1 hybridization buffer (Illumina) and the library diluted post-denaturation to a final loading concentra-tion of 7–8.5 pM. Libraries were sequenced with 10–20% PhiX (Illumina) using the 2 × 300 bp paired-end protocol.

**Independent validation of 16S rRNA sequencing.** To validate the robustness of the microbiota protocol, 10% of DNA samples were shipped to Imperial College London and sequenced according to the methods described above. We included 30 mother–infant pairs per study site in the validation subset and sequenced two infant stools (from weeks 1 and 10/12) and the maternal stool from each pair. Protocol deviations included the use of AMPure XP beads for PCR product pur-ification, fragment analysis via TapeStation D1000 ScreenTape (Agilent), amplicon quantification for individual samples via Qubit dsDNA High Sensitivity Assay Kit (ThermoFisher Scientific), and manual pooling of equimolar quantities. The order of samples was randomised across three PCR plates and the pooled libraries were sequenced with 10–20% PhiX (Illumina) on two 2 × 300 bp paired-end Illumina MiSeq runs (v3 chemistry).

**Bioinformatic processing of sequence data.** Adapters were trimmed from raw sequences using cutadapt version 1.18[41]. Subsequent steps for quality filtering, denoising, merging of paired reads, and chimera removal followed the DADA2 pipeline[42] implemented in QIIME 2 (version 2018.11) with default parameters[43]. Forward reads were truncated to 270 bp and reverse reads to 200 bp to account for the fall in sequencing quality scores towards the end of each read. Sequence data were processed separately for each run, and the feature tables and corresponding sequences then merged. All subsequent steps were performed in the programming language R (version 3.6.1). Taxonomy assignment was performed via the R package dada2 (version 1.14.1) using the RDP naive Bayesian classifier trained on the Silva rRNA database (version 132). Sequence variants were included in the analysis if they were 390–440 bp in length, bacterial, detectable at an abundance of ≥0.1% in at least two samples, and passed frequency-based contamination filtering using the decontam package (version 1.6), which screens sequences that are more abundant in low-concentration samples[44]. Nanodrop concentrations (ng/µl) of extracted DNA served as the basis for contaminant filtering. Samples with at least 25,000 quality-filtered sequences were retained in the final analysis. For visualisation purposes, neighbour-joining trees were constructed at genus level based on JC69 distances, using the sequence of the most abundant RSV as the reference for each genus.

**Statistical analysis.** All analyses were performed in the programming language R. Participants with complete data were included in each analysis. Seroconversion and shedding proportions were compared by country using pairwise two-sided Fisher's exact tests with Benjamini–Hochberg FDR correction. Antibody and inflammatory biomarker concentrations were log transformed and compared by country and shedding subgroup (see Supplementary Fig. 1B) using ANOVA with post-hoc Tukey tests. Maternal breastmilk/serum RV-IgA ratios were calculated using log-transformed antibody concentrations and compared by country using Dunn's test (given their highly skewed distribution). We explored covariates associated with seroconversion, ORV shedding, and neonatal infection status via logistic regres-sion, and those associated with log-transformed post-vaccination RV-IgA via linear regression and Pearson's r with two-sided hypothesis testing. Following these univariate analyses, variables with p values of <0.05 were explored in multivariate models for each ORV outcome. Where multiple correlated variables were eligible for inclusion in multivariate models, we minimised multicollinearity by prioritising variables measured closer to the first dose of ORV (e.g. Shannon index at 6 weeks of life vs 4 weeks of life), maternal antibodies in serum versus breastmilk, and maternal serum RV-IgG versus maternal RV-IgA or pre-vaccination infant RV-IgG. In Indian mother–infant dyads, a correlation network was also calculated between all measured antibody concentrations (RV-IgG and RV-IgA) and rotavirus shedding quantities (reciprocal of qPCR Ct value) using Spearman's rank

correlation coefficients with two-sided hypothesis testing. Clopper–Pearson 95% confidence intervals were used for the reporting of proportions.

Ward's minimum variance hierarchical clustering method was used to visualise all infant and maternal gut microbiota samples based on the presence or absence of common genera (≥1% prevalence across all samples). For cross-sectional analyses of alpha diversity, we used ANOVA with post-hoc Tukey tests (country) or linear regression (ORV outcome and neonatal rotavirus infection status). For cross-sectional analyses of beta diversity, we determined the proportion of variation in genus-level unweighted Bray–Curtis distances associated with country and ORV outcome via PERMANOVA with 999 permutation. A minimum abundance threshold of 0.1% was used when calculating Bray–Curtis distances. To identify discriminant taxa according to country or binary ORV outcomes (seroconversion and shedding), we used two-sided Fisher's exact test (differences in prevalence) and Aldex2 (two-sided Wilcoxon rank-sum test of centred log-ratio transformed sequence counts), classifying taxa as discriminant if they had an FDR-adjusted $p$ value of <0.05 based on either method. Aldex2 was also used to identify taxa correlated with log-transformed post-vaccination RV-IgA (FDR-adjusted $p$ value of <0.05 based on two-sided Spearman's rank test). Taxa were included if they were detected with a prevalence of >5% in at least one of the groups being compared.

We performed an exploratory analysis to identify demographic covariates associated with alpha diversity (via linear regression) or beta diversity (via PERMANOVA of genus-level unweighted Bray–Curtis distances) at the time of the first ORV dose. To account for multiple testing in this exploratory analysis, we report on variables with FDR-adjusted $p$ value of <0.05.

To complement the cross-sectional analyses at each sampling timepoint, we took advantage of the longitudinal study design to explore variation in microbiota diversity and genus abundance across early infancy. Longitudinal mixed-effects models were used to compare Shannon index by country (pairwise comparisons with FDR correction), ORV outcome (seroconversion and dose 1 shedding status), and neonatal rotavirus infection, including week of life as a covariate (to account for age-associated changes in microbiota composition) and study ID as a random effect. For common genera (>20% across infant samples in a given country), we used longitudinal zero-inflated negative binomial models of sequence counts to identify taxa that discriminated infants according to country or ORV outcome, adjusting for age (by including week as a fixed effect) and including study ID as a random effect. A threshold of 20% was selected based on prior validation of this statistical modelling approach in the context of microbiome proportion data[45]. For taxa with a prevalence of >95%, negative binomial models without zero inflation were used.

For technical replicates (including positive controls and samples selected for independent validation), we quantified the proportion of variance explained by sample ID using linear regression (Shannon index, genus abundances) or PERMANOVA with 999 permutations (unweighted Bray–Curtis distances). We also used PERMANOVA to quantify the proportion of variation in microbiota composition associated with sequencing run, stratified by age and country.

**Random Forests**. We applied the Random Forests algorithm in a series of cross-sectional analyses to predict country, seroconversion, or shedding status (classification approach), or post-vaccination log-transformed RV-IgA concentration (regression approach) based on genus or RSV relative abundances. Taxa were included if they were detected with a prevalence of >5% in at least one of the groups being compared. For each analysis, we performed 20 iterations of 5-fold cross-validation. For classification models, we standardised the baseline accuracy at 50% by fitting each iteration of 5-fold cross-validation on a random subset of 50 samples per group (or the number of samples in the minority group if this was <50). Models were excluded if there were <10 samples in the minority group. For regression models, out-of-bag $R^2$ values for predicted vs observed RV-IgA values were determined via linear regression. Mean cross-validation variable importance was determined based on the increase in Gini index (classification) or mean squared error (regression) of out-of-bag sample prediction following random permutation. For Indian infants, we assessed additional models based on a combination of demography/baseline health measurements (18 variables), exposure/antibody data (12 variables, encompassing inflammatory biomarkers, neonatal rotavirus infection status, and maternal antibodies), and genus relative abundances at the time of the first ORV dose (55 variables, encompassing genera present in >5% of samples).

**Sample size**. As reported in the published protocol[37], we calculated that a sample size of 150 infants in India and Malawi would provide 80% power to detect a two-fold higher mean concentration of RV-IgG in infants who fail to seroconvert compared with those who seroconvert (assuming 40% seroconversion in each), while a sample size of 50 in the UK would provide 79% power to detect significant differences in RV-IgA by seroconversion status across the study sites (assuming 95% seroconversion in the UK). These sample sizes would also provide 95% power to detect significant differences in Shannon index according to seroconversion status in India and Malawi. The sample size in India ($n = 307$) exceeded these targets owing to the high recruitment rates in this cohort and the decision to merge the IPV and OPV arms in the final analysis. On the other hand, owing to challenges in recruitment and sample collection over the course of the study, the final

samples size in Malawi ($n = 119$) fell short in these estimates. Since RV-IgG data were not available for the UK and Malawi, our final analyses of maternal antibodies across cohorts focused on RV-IgA.

**Reporting summary**. Further information on research design is available in the Nature Research Reporting Summary linked to this article.

## Data availability
The raw sequence data generated during this study have been deposited in the European Nucleotide Archive under accession code PRJEB38948. Processed data are available on Github (https://github.com/eparker12/RoVI; https://doi.org/10.5281/zenodo.5528337)[46]. Taxonomy assignment was performed using the Silva rRNA database (version 132; https://doi.org/10.5281/zenodo.1172782).

## Code availability
Analysis code generated during this study are available on Github (https://github.com/eparker12/RoVI; https://doi.org/10.5281/zenodo.5528337).

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

## Acknowledgements

We thank all members of the clinical study teams in Vellore, Blantyre, and Liverpool, including Falak Diab, Siobhan Holt, and the research midwives at the Liverpool Women's Hospital; Dawn Redman and the team of research nurses at Alder Hey Children's Hospital; Uma Raman, Charlet, Margaret, Jacklin, and the field research assistants at Christian Medical College, Vellore; and James Tamani, Anna Ainani, Amisa Chisale, Bertha Masamba, Carlo Gondwe and Evelyn Gondwe in Blantyre, Malawi. The UK and Malawi sites were funded by the UK Medical Research Council and the UK Department for International Development (Newton Fund MR/N006259/1). N.A.C. is affiliated to the National Institute for Health Research (NIHR) Health Protection Research Unit in Gastrointestinal Infections at University of Liverpool, in partnership with Public Health England, in collaboration with University of Warwick. N.A.C. is based at The University of Liverpool. The views expressed are those of the author(s) and not necessarily those of the NIHR, the Department of Health and Social Care or Public Health England. K.C.J. is funded by a Wellcome International Training Fellowship (number 201945/Z/16/Z). The site in India was funded by the Government of India's Department of Biotechnology. Richard Eccles, Anita Lucaci, Richard Gregory, John Kenny, and other staff at the Centre for Genomic Research (University of Liverpool) provided valuable support for the 16S microbiota sequencing work, as did Laurence Game at the Imperial BRC Genomics Laboratory. Above all, we are grateful to the families involved in the study.

## Author contributions

Conceptualisation, M.I.G., N.A.C., A.C.D., N.C.G. and G.K.; Methodology, C.B., E.P.K.P., A.C.D., M.I.G., I.P., S.B., and G.K; Software, E.P.K.P.; Validation, E.P.K.P.; Formal analysis, E.P.K.P.; Investigation, C.B., E.P.K.P., N.C.-V., D.H., J.L., S.B., I.P., S.G., S.I., V.K.S., B.B., S.S., B.S.R., J.M., and E.C.; Resources provision, K.N.S., A.G., V.P.V., Q.D., N.C., M.T., and N.A.C.; Data curation, C.B., E.P.K.P., K.N.S., S.V., J.M., and M.I.G.; Writing—original draft, E.P.K.P., C.B., A.C.D., S.B., I.P., A.D., M.I.G., N.C.G., and G.K; Writing—review & editing, B.K.; Visualisation, E.P.K.P; Supervision, N.A.C., A.C.D., K.C.J., N.C.G., B.K., and G.K; Project administration, C.B., M.I.G., I.P., S.B., K.N.S., K.C.J., and G.K.; Funding acquisition, M.I.G. and G.K.

## Competing interests

M.I.G. has received research grants from GSK and Merck, and has provided expert advice to GSK. K.C.J. and N.A.C. have received investigator-initiated research grant support from GSK. N.A.C. has received research grant support and honoraria for participation in rotavirus vaccine data safety monitoring committee meetings from GSK. All other authors declare no competing interests.
