## [Peer Review File · Nature Communications]

Impact of maternal antibodies and microbiota development on the immunogenicity of oral rotavirus vaccine in African, Indian, and European infantsReviewers' Comments:

Reviewer #1:

Remarks to the Author:

This is a valuable addition to the published literature on impaired responses to oral vaccines in many LMICs. The authors, from UK, India and Malawi, have compared responses to oral rotavirus vaccine in children in these three countries.

The study is robust, well-conducted and analysed, and led by a group of investigators who undoubtedly lead the field. The question is one of great importance for global health, as these infections affect millions of children. However, the organisation of text and figures is sub-optimal in this reviewer's opinion. I found myself having to go back and forth between text, figure, figure legend, supplementary figure and supplementary legend for almost every sentence. Where possible, all the information required to support the main conclusions should be in the main text and figures. It should not be necessary to refer to the supplementary material to see how the conclusions were arrived at.

Specific points:

1 The sample set analysed is very complex, with multiple samples collected from different sites, and different vaccines used in India. Great care must be taken to clarify what samples were used at each time. In the first paragraph, the reader is referred to Figure 1A which has a pretty cartoon setting out times of sample collection, but cord blood is omitted and it is not clear throughout the figures whether weeks refers to weeks of life or weeks post-vaccination. Time is usually represented as a horizontal line so a vertical line adds to the sense that the figures are harder to read than they could be.

2 The figures are too crowded and the fonts too small to be read without having to zoom in and out constantly. Legends do not explain the content well (apart from the statistics which were well explained). In Figure 1 (in fact, nowhere that I could find) it was not explained why VP6 and NSP2 were chosen for different assays. Figure 1A does not match the text in the first paragraph; it does not show that virus shedding was near-ubiquitous in the UK. Perhaps that reflects the lack of explanation about whether "weeks" refers to weeks of life or weeks post-vaccination. Clearer legends would help greatly.

3 Some statements could have the results stated simply in the text. For example, "seroconversion was more common after bOPV than tOPV" refers to Table S4 but this could be put in the text rather than making the reader search the supplementary material for a simple fact.

Reviewer #2:

Remarks to the Author:

General Remarks

The study, "Impact of maternal antibodies and microbiota development on the immunogenicity of oral rotavirus vaccine in African, Indian and European infants: a prospective cohort study" by Parker and Bronowski et al is a comprehensive and much-needed addition to the literature addressing the continued poor protection of rotavirus vaccines against serious rotavirus gastroenteritis in low- and middle-income countries (LMIC). The authors are addressing a key unanswered public health question – namely why oral rotavirus vaccines underperform in LMIC. As the authors explain, rotavirus vaccine efficacy is significantly lower in LMIC than in high-income countries. The consequence of this performance gap is that despite widespread uptake of rotavirus vaccines in LMIC and consequent drops in rotavirus mortality, rotavirus still remains the most common cause for a diarrheal hospital admission and death among infants under the age of 2 in these settings. The etiology of this performance gap has eluded scientist, yet remains a critical question in vaccinology as well as public health in global efforts to protect vulnerable infants from life-threatening diarrheal disease.

The authors investigate two hypotheses about the diminished rotavirus vaccine performance in LMIC.

First, whether maternal antibodies may correlate with rotavirus vaccine immunogenicity and second, whether microbiome composition and biomarkers of enteric inflammation may explain performance. In addition, they evaluate numerous sociodemographic risk factors. A great deal of research has already evaluated both these risk factors, including from the authors themselves. This research presented in this article, in which the authors use a prospective and multi-country design, adds to the field in two key ways – first, the authors are able to systematically evaluate the risk factors in relation to one another and second, they can evaluate the risk factors across high- and low-income settings (UK vs Malawi/India) using an identical protocol.

Despite the excellence of their study design, approach, and analysis, the authors' findings are complex and often contradictory. Maternal antibodies are similarly high in India and the UK, yet maternal serum and breast milk anti-RV IgA correlated negatively with vaccine seroconversion in India but not in the UK. Infant microbiome alpha diversity is lower in rotavirus vaccine seroconverters in India and Malawi, but at a country scale, infants in the UK and Malawi have similar alpha diversity. Besides neonatal exposure to rotavirus infection in India, very few risk factors strongly predict rotavirus vaccine seroconversion or shedding, particularly across countries. This can mean one of two things – that the study is not describing/missing the risk factors that really matter in determining rotavirus vaccine performance in LMIC and/or that the vaccine outcomes that the study is using (post-vaccination anti-RV IgA and rotavirus vaccine shedding) are not reflective of true vaccine protection from disease. These possibilities are insufficiently articulated in the study conclusions.

The major limitations of the study and the complex and frequently negative findings should not preclude publication of the results of this rich, layered and detailed study. Yet key limitations should be better addressed/highlighted. These include

- the use of post-vaccination anti-RV IgA and rotavirus vaccine shedding as study outcomes, given how these may be poor correlates of rotavirus vaccine protection. Did the authors evaluate rotavirus vaccine gastroenteritis in the course of the study? Can they present these results?
- The differing sample sizes per country, particularly the low sample size in Malawi and the UK, particularly given the low seroconversion in the UK. The authors do not address sample size calculations in their methodology nor discuss the impact of sample size in interpretation of their study results, despite the sample sizes being smaller for the UK and Malawi than calculated in their published protocol. The significantly lower sample sizes in Malawi and the UK means that a majority of multivariate analyses is performed on the Indian data, skewing results and interpretation and weakening the overarching study aim. Similarly, numerous antibody and EED outcomes are not evaluated in Malawi and India, complicating trans-country interpretation
- Some of the conclusions that the authors draw about impact that maternal antibodies have on rotavirus vaccine shedding. The authors provide insufficient evidence to support the conclusion that maternal antibody may inhibit rotavirus vaccine replication.

The authors use appropriate and valid statistical analyses, with transparent and very detailed workflows including pre-publication of their study protocol, registration in a clinical trial registry, deposition of genomic sequences in public archives, sharing of statistical workflows and detailed methodologies, easily allowing others to to reproduce their work.

Specific remarks

Overall text

Suggest to change uses of 'gut' to faecal microbiome throughout text.

Please change text that describes "ORV replication" and replace it with "ORV shedding" or present data that supports use of the term replication.

Suggest to use term 'markers' of environmental enteric dysfunction

Figures – recommend to not use red and green in figures

Figures – please include sample sizes on all relevant figures

Report FDR, given multiple comparisons in tables

Results

Line 81, Table 1, Baseline characteristics of study cohort – suggest to highlight some key differences by country, namely the high rate of caesarean delivery in India, the low rate of exclusive/partial breastfeeding in the UK and high rate of HIV-exposed in Malawi. (was HIV status not measured in UK?)

Line 80-81 – Table 1

Were there significant differences in week of gestation by country? If so, please report.

Figure 1A – rather complex graphic, recommend to place horizontally so as to orient the reader in time

Sampling – please indicate in text the relationship of sampling to vaccination (pre or post and with what window)

ORV replication and immunogenicity

Line 90 – ORV replication and immunogenicity
remove 'replication' from title and replace with shedding

Line 92 – ..(44%) continuing to shed after 1 month

Line 93-94 ... continued shedding after 1 month

Please clarify in both these texts that this was pre vaccine dose 2 and not post-vaccine dose 2?

Sampling numbers appear to change over time, please report sample size for all figures.

Lines 95-97: Shedding following at least one dose was observed

Authors may be presenting too many vaccine outcomes here, suggest to select one shedding outcome and report consistently. (e.g. either after dose one, or after at least one dose)

Figure 1B

please indicate when shedding was measured (1 week post dose?), and please define VP6 and NSP2 for the reader. Please adjust box plots to indicate sample size.

Line 99 - Seroconversion was observed

While authors describe this in the following paragraph, it would help to indicate proportion of infants per cohort that are seropositive prior to vaccination. While defined in methods, suggest to include a definition of seroconversion here for the reader.

Figure 1D –

The reviewer does not see the added value of this figure, which combines seroconversion and shedding data and suggests to remove it. This is because in figures 1B-C, the authors show the underlying complexity and divergence of the shedding and seroconversion data, particularly the high numbers of infants in India with pre-existing/neonatal infection which likely reduced post-ORV shedding but also perhaps increased seroconversion. Rather than clarifying understanding of immune response to ORV based on immunogenicity and shedding, the combination figure confounds it

Line 103-5 –Overall, at least one indicator of ORV response

Suggest to remove this section along with figure 1D as described above

Neonatal rotavirus infection

In this paragraph, the authors jump between evaluated populations – sometimes referring to infants

with high pre-vaccination IgA, sometimes to 'previously infected', which seems to imply infants who are seropositive and/or neonatal shedders. This is unnecessarily confusing and suggest to restrict this paragraph to one analyzed population only. Additionally, the authors use the terminology "pre-vaccination rotavirus exposure" in figure S1 but "previously infected" here and in figure 1E- and later in the text use neo + and -. Please define the terminology, harmonize and utilize one definition throughout text

Line 118 – Prior infection was associated with a reduced likelihood of shedding ORV
Given reference to only seroconversion in line 116, please define "prior infection" are these seropositive infants or seropositive +/- pre-dose 1 shedders? See also above

Line 119-120 – "by contrast, pre-vaccination infection did not influence the likelihood of serconversion"
the authors have two definitions of seroconversion (for >20 IU for seronegative infants and 4-fold titer increase for seropositive) – please clarify the baseline population and what was analysed here.

Line 119-121 – By contrast pre-vaccination infection did not influence the likelihood of seroconversion...and where ORV replication was observed...
again, this sentence is confusing. Separate these two findings if the populations that were being analyzed are not the same. It is not clear from the text what the second part of this sentence is describing (only from the figure) "where ORV replication was observed" – the figure suggests the authors mean post dose 1 . Also remove term 'replication'.

Also – suggest that authors provide figure that is identical to figure 1E, but shows proportion seroconversion (instead of using figure S1B)

Figure 1E – define "pre-ORV infection" and "any vaccine shedding"

Line 131-2 – suggest to help the reader by explaining the significance of the dry season in India on RV circulation

Line 133 – suggest to add "were apparent in India"

Line 134 – add "in India but not Malawi and the UK" after "first ORV dose"

Line 136 – add "in India"

Line 138 – suggest to change "strongly" to "significantly"

Line 139 - suggest to add "(Table S5)"

Maternal antibodies

While both are statistically significant, the authors emphasize transplacental over breastmilk antibodies in this section. Suggest to reduce interpretation in results section and/or provide a more balanced interpretation of relative importance of IgG transplacental transfer vs breastmilk IgA.

Line 152, remove "this may reflect a lack of statistical power" and move to discussion. Please also include sample size for all figures

Line 154-5 - "Serum IgA is not transmitted efficiently across the placenta and is therefore unlikely to directly influence ORV". What about the correlation between serum IgA and breastmilk IgA? Breastmilk IgA may "directly influence ORV". Suggest to modify sentence to be more specific and/or move to discussion.

Figure 2B – use uniform terminology for previously infected cohort (neo+/- vs infected in other figures)

Line 157-159 – "maternal serum RV-IgA levels were negatively correlated with ORV shedding in both India and Malawi (Figure 2C and Table S3)".

Figure 2C shows a significant negative correlation between maternal IgA and shedding in India –

primarily driven by the neo+ group, however Table S3 shows no significant correlation for maternal serum IgA and shedding in Malawi after dose 1 or after either dose. Please support conclusion with significant results or remove.

Line 160 - It appears that maternal antibodies inhibited both neonatal antibody anti-RV responses and post-vaccination anti-ORV responses, but not neonatal RV shedding in the Indian infants. How do the authors interpret this?

Paragraph 166, authors suggest that breastmilk IgA may be less inhibitory than transplacental antibodies, however maternal serum IgA and breastmilk IgA both correlate negatively with shedding whereas maternal IgG and cord blood IgG do not. This data suggest that there may be divergent mechanisms driving correlations between breastfeeding and antibody and shedding responses. Suggest to modify sentence on line 169 or move to discussion.

Inflammatory biomarkers

Why were the same inflammatory biomarkers not measured across all countries?

Geographic differences in microbiota development

Line 180-1 – “We obtained high quality faecal microbiota profiles in 2,086 samples”

Please provide total number of samples sequenced (2,138?)

Line 188-191 Why do authors only name the genera enriched in India and UK but not in Malawi (suggest to either remove from text or add major genera enriched in Malawi)

Figure 3C report significance by country

Sentence 198-199 “Indeed, simple cross-sectional comparisons of taxon prevalence were adept at selecting differential microbiota colonization patterns...”

what extra information does this provide beyond what is additionally presented in figure S5A and B? Suggest to streamline and remove.

Line 204-5 “Interestingly, stratified analyses”

add “in India”

Microbiota composition versus ORV response

Line 201 – the authors describe an interesting negative correlation between microbiota alpha diversity and ORV immunogenicity that appears to be the most pronounced in the non-rotavirus exposed group in India and the time of first ORV vaccination in Malawi. It is surprising, given the significance in alpha diversity, that there are no corresponding differences in taxonomic composition/beta diversity according to ORV response. The authors have used Random Forest models for their cross-sectional analysis and also present a longitudinal analysis in figure 4E. Please provide a rationale for why these methodologies were used. Given the differences in alpha diversity, the reviewer requests a more detailed analysis of beta diversity. The reviewer requests the following

- Analysis of differences in taxonomic composition according to ORV response at the sequence variant level (github analysis suggests both genus and sequence variant level analyses were done)
- Given the difference in alpha diversity by seroconversion for the neo- population, the reviewer requests an analysis of difference in alpha and beta microbiome composition between neo + and neo – infants in India. Additionally, it is striking that, only for the infants in the neo- group, infants are significantly more likely to seroconvert if they have had a caesarian section (particularly as infants with caesarian section are known to have slower colonization rates). Please also compare the alpha and beta microbiota diversity of infants with and without a cesarian section in this group to evaluate if delivery mode may be impacting/confounding alpha diversity/microbiome maturation and seroconversion in this subgroup.
- Alpha diversity is lower in UK than Malawi but similar between UK and India. Is UK alpha diversity

lower than Indian neo- alpha diversity? In other words, is there a consistent trend between low alpha diversity and increased seroconversion across countries and within countries when the neo + group is removed?

- Please perform a complementary differential abundance analysis to the cross-sectional random forest models, such as DESeq or equivalent that is tailored to microbial genomic data, to orthogonally evaluate differential abundance by seroconversion and shedding per time point, particularly for neo-group in India and at time of first vaccination for infants in Malawi

- Please evaluate if there are significant differences in microbiome composition according to demographic outcomes that correlate with ORV immunity and could be confounding microbiome analyses– including delivery in a tertiary care facility, delivery by cesarian section, exclusive breast feeding, height for age

- Figure S7 – figure E, what is the asterix referring to? figure D appears incorrectly labeled in text (should be F).

- Line 218 – what do the authors mean with “microbial exposure” and how is this supported by their findings? Suggest to remove. Did the authors consider calculating a Maturation index for microbiome development to support this notion of colonization?

Multivariate analysis

Please indicate in figure 5 and in the title of its legend that this analysis is only of Indian infants.

Authors include pre-vaccination microbiota composition in the multivariate analysis. Please describe in more detail what criteria were used for the 126 genus abundances.

Line 230 - the authors should better describe the results presented in figure 5 in the text – they now describe them without contextualizing them to the cohorts (all India/neo +/-neo-) analyzed. Suggest to address: Pre-vaccination rotavirus infection is the most important predictor of IgA concentration for the Indian population as a whole and maternal IgG concentration in the exposed cohort predicts IgA concentration. However, for non-exposed infants, the effect sizes of variables is considerably diminished and multiple microbiota and antibody variables correlated with IgA concentration.

A shortcoming of these regressions is the repeated use of post-vaccination IgA concentration as outcome– which this paper itself shows is an imperfect correlate of protection. Were the same outcomes found when correlating with rotavirus vaccine shedding?

Discussion

The authors have analyzed and collated a dense and impressive degree of geographic, demographic, immune, and microbiome risk factors for rotavirus vaccine performance. This makes the discussion especially important in leading the reader through the significant and most important non-significant findings. However, in the discussion, the authors are summarizing their findings as according to their pre-existing sections and do not prioritize their findings or offer the reader an overarching interpretation of their work. Foremost among the open questions is if the authors believe the risk factors they evaluated and their study findings (singularly or in combination) adequately explain the diminished protection rotavirus vaccines afford infants in LMIC. Did they meet their study aim? And secondly, how their findings may have shifted understanding of key questions around rotavirus vaccine performance – eg. that enteropathy/low diverse microbiota may not be driving poor performance; the potential importance of early (neonatal rota) viral exposures; the complexity of maternal antibody responses and vaccine performance. And while they list future directions, what work deserves priority in the future?

Line 262-3 We advocate the further use of vaccine virus replication as an adjunctive measure of oRV response in future trials

While the reviewer agrees that rotavirus vaccine shedding may be a better correlate of protection than post-vaccination IgA, this work has not evaluated vaccine shedding in relation to clinical protection

from severe rotavirus gastroenteritis, and therefore it is unclear on what basis the authors are advocating this standpoint. Additionally, there are numerous technical difficulties in using shedding as an outcome which also deserve to be addressed – if serial samples are not obtained, which day post-vaccination should be used? What is the likelihood that shedding can be missed?

Lines 277-8 – “we observed a reduction in both shedding and immunogenicity, suggesting that the effect of maternal antibodies is mediated in part through a reduction in ORV replication efficiency” the reviewer disagrees that authors have demonstrated that maternal antibodies had a clear effect on shedding alongside immunogenicity. The authors demonstrate a significant negative correlation between maternal antibodies and post-ORV IgA. However, the correlation with shedding is only for maternal IgA and only in the neo + group in India – not in any other cohort or subgroup. This makes extrapolation from maternal antibodies to shedding purely speculative. They have also not demonstrated an effect on replication as they have not measured replication. That the effect is only seen in the neo + group suggests instead that a negative correlation of maternal antibodies on shedding in exposed infants may be mediated instead through an effect on infant IgA/IgG pre-ORV. Please address and modify rest of paragraph accordingly.

Line 281 – relevance of passively acquired IgG antibodies and lentiviruses is unclear, suggest to remove

Line 283 – the authors have not shown that there is a leaky mucosal barrier or early-life enteropathogen exposure in their populations – rather their results suggest the opposite, that EED biomarkers do not differ significantly by ORV response. Please address and suggest to remove speculation not in line with findings.

Line 302 – suggest to change language “...oral vaccine failure and/or that the condition...(quite possibly both)”

Line 303 – reviewer suggests that authors also address microbiota differences within country (e.g. was the neo + and neo – group significantly different in alpha and beta diversity)

Line 310 please address that alpha diversity results found within countries showing that higher alpha diversity correlates with lower immunogenicity are not found between countries (where one would expect lower diversity in UK compared to both India/Malawi)

Line 315 – address possible correlation between cesarian section and microbiome diversity in India.

Line 315 –For example, microbiota richness may act as a proxy for early life exposures that shape ORV outcome, such as non-polio enteroviruses the reviewer does not completely follow the authors’ rationale here – low diversity correlates with increased seroconversion. The reviewer would assume viral infection/non-polio enterovirus exposure would result in reduced microbiota diversity rather than increased diversity (as has been shown with numerous enteric viral and bacterial diarrheal diseases). Then NPEV would be positively correlated with seroconversion. Whereas in the literature NPEV and OPV are negatively correlated with seroconversion (Taniuchi, Vaccine 2016).

Line 317 – The fact that ORV immunogenicity was impaired among Indian infants in households without access to treated water would support this notion However, other studies have shown that infants without access to treated water and poor living conditions generally have lower microbial diversity and microbiome “immaturity” (Subramanian, Nature 2014), yet the current study has found that infants with lower diversity have higher seroconversion. Please address.

Line 323-4 – a clear understanding of the signature of ‘healthy’ microbiota development in different

settings is lacking

Please revise to address growing body of literature on microbiome diversity and malnutrition among young children in low-income settings such as Malawi and Bangladesh (e.g. Smith M, Science 2013; Chen R, NEJM, 2020; Gehrig J, Science 2020)

Line 337 – this study advances our understanding of the potential mechanisms influencing ORV response in several ways

The reviewer agrees with this statement and this work is a very comprehensive evaluation of possible known risk factors for impaired rotavirus vaccine performance in LMIC. However, the work has not yet uncovered clear answers about why rotavirus vaccines are insufficiently immunogenic in these populations. The reviewer requests that the authors speculate about whether risk factors are missing or if they believe their work demonstrates that a combination of factors is driving the observed differences in vaccine efficacy between high and LMIC or if they used the wrong correlates of vaccine protection.

Line 341 – the reviewer disagrees with the authors conclusion that they have shown that maternal antibodies reduce ORV replication. See comments to lines 277-8 above, in addition they have not evaluated replication, only looked at the presence or absence of ORV shedding. Suggest to remove or provide evidence to support this claim.

Line 341-44 – neonatal rotavirus exposure in India is one of the most striking observations in this paper, with by far the largest reported effect size (Figure 5B). Given this was one of the most significant finding in the manuscript, this deserves far more weight in the discussions. Please address.

Line 345 – the reviewer is not sure what the authors mean to suggest with 'high microbial exposure'. Do the authors mean enteropathogen exposure? More rapid colonization? Higher alpha diversity? Please specify and address and support the corresponding hypothesis in discussion prior to using this as a concluding statement.

Methods

Study design

Please describe when shedding samples were collected in relation to vaccination (days) and over what range of time collection were considered acceptable for inclusion.

Line 550 Why was cord blood and maternal serum collected but no maternal serum IgG, cord blood IgG reported for Malawi and UK? Please explain or include this data

Line 601 What was the range of the shedding Ct values used for correlation analyses.

Line 640 Why were MPO and alpha1AT not performed for UK infants?

Statistical analysis

Provide a rationale for the study sample size including a rationale for why the sample size is so different across the three sites. Please also address this in limitations of the study in the discussion. In the published study protocol (Sindu BMJ, 2017), the sample size was estimated to require 150 mother infant pairs in India and Malawi and 55 in the UK for infant seroconversion outcomes. Why was sample size so much lower in Malawi and what impact may this have had on study results? Also discuss the lower seroconversion than expected in the UK and how this relates to the power calculations that were made for the UK which assumed > 95% seroconversion.

Line 741-2 antibody concentration is assumed to be linear, why was Spearman used over Pearson's?

741-2 authors used 1/Ct value for shedding – however what was considered a negative Ct value and

how was quantification validated? Shedding was evaluated twice, how did the authors use multiple shedding results in their correlation analyses?

We are grateful for the constructive comments of the two reviewers, and have provided point by point responses to each comment below. Line numbers refer to the version of the revised manuscript without tracked changes.

Reviewer #1 (Remarks to the Author):

This is a valuable addition to the published literature on impaired responses to oral vaccines in many LMICs. The authors, from UK, India and Malawi, have compared responses to oral rotavirus vaccine in children in these three countries.

The study is robust, well-conducted and analysed, and led by a group of investigators who undoubtedly lead the field. The question is one of great importance for global health, as these infections affect millions of children. However, the organisation of text and figures is sub-optimal in this reviewer's opinion. I found myself having to go back and forth between text, figure, figure legend, supplementary figure and supplementary legend for almost every sentence. Where possible, all the information required to support the main conclusions should be in the main text and figures. It should not be necessary to refer to the supplementary material to see how the conclusions were arrived at.

Author response: We thank the reviewer for their constructive response to the paper. The comments on manuscript organisation are well taken, and we have modified the text and figures throughout the paper to address these concerns. Key changes are detailed in our responses to the specific comments from both reviewers below, but include:

- Simpler figure formats in which charts and font sizes are larger, and sample sizes are presented below every panel.
- Expanded figure legends which provide key methodological details, thereby diminishing the need to refer to supplementary text (see response to specific point 1.2 for details).
- We are more explicit in the main text about exactly which sampling timepoint is being referred to. For example, at line 114: *"24/54 (44%) continuing to shed immediately prior to the second dose (4 weeks after dose 1; Figure S1A)."*
- Where possible, we have brought key results into the main text to diminish the need to refer to the supplementary materials (see response to specific point 1.3 for details).
- Key methodological details have been moved from the supplementary materials to the main text. For example, at line 86: *"Starting in the first week of life, six longitudinal stool samples were collected from each infant and assayed for rotavirus shedding, with samples collected 1 week after each ORV dose providing an indicator of vaccine virus take. As a proxy for bacterial microbiota development in the infant gut, we sequenced the V3-V4 region of the 16S rRNA gene in stool samples collected at 1 and 4 weeks of age, and before each ORV dose."* Similarly, the paragraph describing the sequentially recruited IPV arm in India is now been moved to the Results rather than the Methods (line 101 onwards).
- To simplify **Figure 5**, panel C is now presented separately as **Table 2**.

Specific points:

1.1) The sample set analysed is very complex, with multiple samples collected from different sites, and different vaccines used in India. Great care must be taken to clarify what samples were used at each time. In the first paragraph, the reader is referred to Figure 1A which has a pretty cartoon setting out times of sample collection, but cord blood is omitted and it is not clear throughout the figures whether weeks refers to weeks of life or weeks post-vaccination. Time is usually represented as a horizontal line so a vertical line adds to the sense that the figures are harder to read than they could be.

Author response: Thanks for this valuable comment. As suggested:

- We have reoriented **Figure 1A** to place time on the horizontal axis, added cord blood, and clarified that this axis refers to 'week of life'. We have also modified the figure to a simpler heatmap format.
- In all later figures, we now refer to 'week of life' when specifying the sample timing, and clarify the precise number of samples involved in each figure panel.

1.2) The figures are too crowded and the fonts too small to be read without having to zoom in and out constantly. Legends do not explain the content well (apart from the statistics which were well explained). In Figure 1 (in fact, nowhere that I could find) it was not explained why VP6 and NSP2 were chosen for different assays. Figure 1A does not match the text in the first paragraph; it does not show that virus shedding was near-ubiquitous in the UK. Perhaps that reflects the lack of explanation about whether "weeks" refers to weeks of life or weeks post-vaccination. Clearer legends would help greatly.

Author response: We have made the following changes to address these comments:

- The comment on font size is well taken. We have increased the font size in figures throughout the manuscript and supplement.
- Where possible, we have reduced the number of panels so as to simplify the presented data. For example, **Figure 1D** has been removed and **Figure 1E** shifted to the supplement, providing more space for the remaining panels.
- We now consistently refer to 'week of life' throughout the figures, also clarifying timing relative to vaccination where relevant. This helps draw attention to the fact that the 'near-ubiquitous' shedding in **Figure 1B** (middle and right panels) is observed in samples obtained 1 week after ORV.
- In **Figure 1**, we have added the following text to clarify the selected PCR targets (VP6 and NSP2) in the legend: "Rotavirus shedding was detected via quantitative PCR using a pan-rotavirus assay targeting the VP6 gene of group A rotaviruses (week of life 1) and an assay for vaccine virus shedding targeting the Rotarix NSP2 gene (1 week after each dose)." We thank the reviewer for highlighting this omission.
- We clarify the number of samples involved in all figure panels.
- Legends have been expanded to provide clearer information on the data being presented. For example, the legend to Figure 1 now reads as follows:

"Figure 1. Study Design and Oral Rotavirus Vaccine Response. (A) Study design. The final study population comprised 307 infants in India, 119 in Malawi, and 60 in the UK. (B and C) Geographic differences in (B) rotavirus shedding and (C) ORV immunogenicity. Rotavirus shedding was detected via quantitative PCR using a pan-rotavirus assay targeting the VP6 gene of group A rotaviruses (week of life 1) and an assay for vaccine virus shedding targeting the Rotarix NSP2 gene (1 week after each dose). Seroconversion was defined as detection of RV-IgA at ≥ 20 IU/ml post-vaccination among infants who were seronegative at baseline or a 4-fold increase in RV-IgA concentration among infants who were seropositive at baseline. Error bars represent Clopper-Pearson 95% confidence intervals. Groups were compared by Fisher's exact test with FDR correction (binary outcomes) or ANOVA with post-hoc Tukey tests (continuous outcomes). The dotted lines at 20 IU/ml indicate the standard cut-off for RV-IgA seropositivity."

Equivalent explanatory text has been added to the other figure legends, as detailed in the tracked changes.

1.3) Some statements could have the results stated simply in the text. For example, "seroconversion was more common after bOPV than tOPV" refers to Table S4 but this could be put in the text rather than making the reader search the supplementary material for a simple fact.

Author response: We have updated the manuscript as suggested. Specific examples are as follows:

- Line 157: "[Seroconversion] was less common in infants who received tOPV-containing than bOPV-only schedules (RRs of 0.56 [0.29–0.95] for tOPV-only and 0.56 [0.29–0.95] for mixed tOPV/bOPV)..."
- Line 185: "Second, maternal serum RV-IgA levels were negatively correlated with ORV shedding after dose 1 in India (RR 0.68 [0.53–0.85]; **Figure 2C**), particularly among infants with neonatal infection (RR 0.50 [0.31–0.75]), and a similar trend was evident in Malawi (RR 0.85 [0.67–1.00]; **Table S3**)."
- Line 199: "Breastmilk RV-IgA was negatively correlated with ORV shedding after dose 1 in India (RR 0.83 [0.69–0.98]; **Table S3**) but not Malawi (RR 0.91 [0.58–1.23])."

We feel that these changes have significantly improved the structure and coherence of the presented data, and thank the reviewer for their valuable suggestions.

Reviewer #2 (Remarks to the Author):

General Remarks

The study, "Impact of maternal antibodies and microbiota development on the immunogenicity of oral rotavirus vaccine in African, Indian and European infants: a prospective cohort study" by Parker and Bronowski et al is a comprehensive and much-needed addition to the literature addressing the continued poor protection of rotavirus vaccines against serious rotavirus gastroenteritis in low- and middle-income countries (LMIC). The authors are addressing a key unanswered public health question – namely why oral rotavirus vaccines underperform in LMIC. As the authors explain, rotavirus vaccine efficacy is significantly lower in LMIC than in high-income countries. The consequence of this performance gap is that despite widespread uptake of rotavirus vaccines in LMIC and consequent drops in rotavirus mortality, rotavirus still remains the most common cause for a diarrheal hospital admission and death among infants under the age of 2 in these settings. The etiology of this performance gap has eluded scientist, yet remains a critical question in vaccinology as well as public health in global efforts to protect vulnerable infants from life-threatening diarrheal disease.

The authors investigate two hypotheses about the diminished rotavirus vaccine performance in LMIC. First, whether maternal antibodies may correlate with rotavirus vaccine immunogenicity and second, whether microbiome composition and biomarkers of enteric inflammation may explain performance. In addition, they evaluate numerous sociodemographic risk factors. A great deal of research has already evaluated both these risk factors, including from the authors themselves. This research presented in this article, in which the authors use a prospective and multi-country design, adds to the field in two key ways – first, the authors are able to systematically evaluate the risk factors in relation to one another and second, they can evaluate the risk factors across high- and low-income settings (UK vs Malawi/India) using an identical protocol.

Despite the excellence of their study design, approach, and analysis, the authors' findings are complex and often contradictory. Maternal antibodies are similarly high in India and the UK, yet maternal serum and breast milk anti-RV IgA correlated negatively with vaccine seroconversion in India but not in the UK. Infant microbiome alpha diversity is lower in rotavirus vaccine seroconverters in India and Malawi, but at a country scale, infants in the UK and Malawi have similar alpha diversity. Besides neonatal exposure to rotavirus infection in India, very few risk factors strongly predict rotavirus vaccine seroconversion or shedding, particularly across countries. This can mean one of two things – that the study is not describing/missing the risk factors that really matter in determining rotavirus vaccine performance in LMIC and/or that the vaccine outcomes that the study is using (post-vaccination anti-RV IgA and rotavirus vaccine shedding) are not reflective of true vaccine protection from disease. These possibilities are insufficiently articulated in the study conclusions.

The major limitations of the study and the complex and frequently negative findings should not preclude publication of the results of this rich, layered and detailed study. Yet key limitations should be better addressed/highlighted. These include:

- the use of post-vaccination anti-RV IgA and rotavirus vaccine shedding as study outcomes, given how these may be poor correlates of rotavirus vaccine protection. Did the authors evaluate rotavirus vaccine gastroenteritis in the course of the study? Can they present these results?
- The differing sample sizes per country, particularly the low sample size in Malawi and the UK, particularly given the low seroconversion in the UK. The authors do not address sample size calculations in their methodology nor discuss the impact of sample size in interpretation of their study results, despite the sample sizes being smaller for the UK and Malawi than calculated in their published protocol. The significantly lower sample sizes in Malawi and the UK means that a majority of multivariate analyses is performed on the Indian data, skewing results and interpretation and weakening the overarching study aim. Similarly, numerous antibody and EED outcomes are not evaluated in Malawi and India, complicating trans-country interpretation

- Some of the conclusions that the authors draw about impact that maternal antibodies have on rotavirus vaccine shedding. The authors provide insufficient evidence to support the conclusion that maternal antibody may inhibit rotavirus vaccine replication.

The authors use appropriate and valid statistical analyses, with transparent and very detailed workflows including pre-publication of their study protocol, registration in a clinical trial registry, deposition of genomic sequences in public archives, sharing of statistical workflows and detailed methodologies, easily allowing others to reproduce their work.

Author response: We would like to thank the reviewer for their detailed consideration of the manuscript. Their thoughtful input has helped make it a more robust and substantial piece of work. We have provided detailed responses to each of the specific critiques below. With regard to the three bullet points above:

- We now discuss the lack of longer-term follow-up for rotavirus-associated gastroenteritis under the study limitations as follows (line 418): *"We compensated for this by considering multiple endpoints, including seroconversion and post-vaccination shedding, though longer-term follow-up for rotavirus-associated gastroenteritis was beyond the scope of the study."*
- We now explicitly discuss sample size calculations in the Methods section. Relevant deviations are outlined under the study limitations. For details, refer to specific comment 2.76 below.
- We have toned down the conclusions regarding the link between maternal antibodies and shedding, as detailed in comments 2.58, 2.59, and 2.60 below.

Furthermore, we have substantially expanded our discussion regarding the limited extent to which the measured risk factors predicted ORV failure in these cohorts, and elaborated on possible reasons for this (comment 2.56 below).

Specific remarks

Overall text

2.1) Suggest to change uses of 'gut' to faecal microbiome throughout text.

Author response: We have opted to keep with the term 'gut microbiota' as is pervasive in the microbiome literature. However, the reviewer's point is well taken and we have added the following sentence to clarify the fact that these samples should be viewed as an imperfect proxy (line 89): *"As a proxy for bacterial microbiota development in the infant gut, we sequenced the V3-V4 region of the 16S rRNA gene in stool samples collected at 1 and 4 weeks of age, and before each ORV dose."*

2.2) Please change text that describes "ORV replication" and replace it with "ORV shedding" or present data that supports use of the term replication.

Author response: Modified as suggested throughout the manuscript. For example, "*ORV shedding and immunogenicity*" is the updated subheading on line 113.

2.3) Suggest to use term 'markers' of environmental enteric dysfunction

Author response: Modified as suggested throughout the manuscript. For example, "*EED markers were measured in serum and/or stool samples*" on line 91.

2.4) Figures – recommend to not use red and green in figures

Author response: Thanks for pointing out this oversight. We have amended all figures to colour-blind-accessible palettes.

2.5) Figures – please include sample sizes on all relevant figures

Author response: Modified as suggested for all main and supplementary Figures. We have also removed **Table S1**, where sample sizes were previously reported.

2.6) Report FDR, given multiple comparisons in tables

Author response: We report FDR p values for each cross-sectional analysis of genera and RSVs in relation to country or ORV outcome. We also report FDR p values for the exploratory analysis of cofactors associated with microbiota composition (**Figure S6**).

The primary analyses of maternal antibodies, EED, and microbiota diversity were specified *a priori* and therefore our use of unadjusted p values is appropriate. We also include a selected number of demographic variables on an exploratory basis, also with unadjusted p values. In all cases where significant associations with ORV outcome were found, we report both univariate outcomes and multivariate outcomes adjusting for other significant covariates (**Tables S2 to S4**). We therefore feel that the approach is appropriate for an exploratory analysis of cofactors associated with ORV outcome, and is one consistent with recent work by others in this area (e.g., Church et al, *Vaccine* 38:2870-2878).

Results

2.7) Line 81, Table 1, Baseline characteristics of study cohort – suggest to highlight some key differences by country, namely the high rate of caesarean delivery in India, the low rate of exclusive/partial breastfeeding in the UK and high rate of HIV-exposed in Malawi. (was HIV status not measured in UK?)

Author response: We agree that it is useful to highlight key differences between the study populations at the outset of the Results, and have therefore added the following paragraph to address this (line 95):

“While the pre- and post-parturition conditions of these disparate cohorts are innumerable, several distinguishing features are highlighted in Table 1. Infants in the UK were characterised by a higher birthweight and a greater prevalence of formula feeding (though >75% were partially or exclusively breastfed). Elective caesarean was an exclusion criterion for the UK but was the mode of delivery for 70/307 (23%) infants in India. HIV exposure was common among infants in Malawi (27/119 [23%]).”

2.8) Line 80-81 – Table 1: Were there significant differences in week of gestation by country? If so, please report.

Author response: Gestational age was only recorded for Indian infants; a comparison of this variable by country is therefore beyond the scope of the present study. As noted on line 703, pre-term birth (<34 weeks gestation) was among the study’s exclusion criteria.

2.9) Figure 1A – rather complex graphic, recommend to place horizontally so as to orient the reader in time

Author response: As suggested, we have reoriented **Figure 1A** to place week of life on the horizontal axis. We have also amended the figure to a simpler heatmap format.

2.10) Sampling – please indicate in text the relationship of sampling to vaccination (pre or post and with what window)

Author response: We have expanded the description of the study design to clarify sample timing relative to vaccination as follows (line 83):

“We measured rotavirus-specific IgA (RV-IgA) in maternal blood, cord blood, and breastmilk samples collected during or in the week after delivery, and in infant blood samples collected pre- and post-vaccination (4 weeks after dose 2) ... Starting in the first week of life, six longitudinal stool samples were collected from each infant and assayed for rotavirus shedding, with samples collected 1 week after each ORV dose providing an indicator of vaccine virus take. As a proxy for bacterial microbiota development in the infant gut, we sequenced the V3–V4 region of the 16S rRNA gene in stool samples collected at 1 and 4 weeks of age, and before each ORV dose. EED markers were measured in serum and/or stool samples collected before each vaccine dose (Figure 1A).”

ORV replication and immunogenicity

2.11) Line 90 – ORV replication and immunogenicity: remove ‘replication’ from title and replace with shedding

Author response: Modified as suggested.

2.12) Line 92 – ..(44%) continuing to shed after 1 month
Line 93-94 ... continued shedding after 1 month

Please clarify in both these texts that this was pre vaccine dose 2 and not post-vaccine dose 2?

Author response: Clarified as follows (line 114): *“continuing to shed immediately prior to the second dose (4 weeks after dose 1; Figure S1A). By contrast, dose 1 shedding was detected in 82/305 (27%) infants in India and 56/101 (55%) in Malawi (Figure 1B), and continued shedding prior to the second dose was much rarer in these cohorts (Figure S1A).”*

2.13) Sampling numbers appear to change over time, please report sample size for all figures.

Author response: As suggested, sample sizes are now reported below all figure panels.

2.14) Lines 95-97: Shedding following at least one dose was observed

Authors may be presenting too many vaccine outcomes here, suggest to select one shedding outcome and report consistently. (e.g. either after dose one, or after at least one dose)

Author response: This point is well taken. We have cut the line specifying shedding rates following dose 2 ("*ORV shedding after the second dose...*"), but have opted to keep data on dose 1 shedding and cumulative shedding (after either dose) in **Figure 1B** so as to clarify the overall shedding rates in each cohort. As per the reviewer's suggestion, we now restrict later analyses (of cofactors associated with ORV response) to dose 1 shedding.

2.15) Figure 1B

please indicate when shedding was measured (1 week post dose?), and please define VP6 and NSP2 for the reader. Please adjust box plots to indicate sample size.

Author response: Modified as suggested. Specifically, sample timings ("*1 week after dose 1*" and "*1 week after either dose*") have been added to the plot headers in **Figure 1B**, and the following text regarding PCR targets has been added to the figure legend: "*Rotavirus shedding was detected via quantitative PCR using a pan-rotavirus assay targeting the VP6 gene of group A rotaviruses (week of life 1) and an assay for vaccine virus shedding targeting the Rotarix NSP2 gene (1 week after each dose).*" Since the gene targets are now clarified in the legend, we have removed this information from the plot headers.

2.16) Line 99 - Seroconversion was observed

While authors describe this in the following paragraph, it would help to indicate proportion of infants per cohort that are seropositive prior to vaccination. While defined in methods, suggest to include a definition of seroconversion here for the reader.

Author response: Modified as suggested (line 121):

"We observed similar geographic discrepancies in ORV immunogenicity. Baseline seropositivity was common in India (99/305 [32%] compared to <5% in the UK and Malawi) due to high rates of neonatal rotavirus exposure in this cohort (see below). Seroconversion, defined as detection of RV-IgA at ≥ 20 IU/ml in previously seronegative infants or a 4-fold increase in RV-IgA concentration among infants who were seropositive at baseline..."

2.17) Figure 1D -

The reviewer does not see the added value of this figure, which combines seroconversion and shedding data and suggests to remove it. This is because in figures 1B-C, the authors show the underlying complexity and divergence of the shedding and seroconversion data, particularly the high numbers of infants in India with pre-existing/neonatal infection which likely reduced post-ORV shedding but also perhaps increased seroconversion. Rather than clarifying understanding of immune response to ORV based on immunogenicity and shedding, the combination figure confounds it

Author response: The reviewer's point is well taken. As suggested, we have removed **Figure 1D**.

2.18) Line 103-5 - Overall, at least one indicator of ORV response

Suggest to remove this section along with figure 1D as described above

Author response: As suggested, this sentence has been removed.

Neonatal rotavirus infection

2.19) In this paragraph, the authors jump between evaluated populations - sometimes referring to infants with high pre-vaccination IgA, sometimes to 'previously infected', which seems to imply infants who are seropositive and/or neonatal shedders. This is unnecessarily confusing and suggest

to restrict this paragraph to one analyzed population only. Additionally, the authors use the terminology “pre-vaccination rotavirus exposure” in figure S1 but “previously infected” here and in figure 1E- and later in the text use neo + and -. Please define the terminology, harmonize and utilize one definition throughout text

Author response: Thank you for this valuable comment. As suggested, we now establish a single, harmonised definition of neonatal infection at the outset of this paragraph, and refer consistently to ‘neonatal infection’ thereafter. Specifically, at line 133:

*“A distinct feature among Indian infants was the high rate of neonatal rotavirus infection, which we defined as the detection of wild-type rotavirus shedding in week of life 1 (**Figure 1B**) or baseline seropositivity (pre-vaccination RV-IgA ≥ 20 IU/ml). This was observed in 166/304 (55%) infants in India, whereas the corresponding rates were 10/90 (11%) in Malawi and 2/54 (4%) in the UK. Neonatal infection was more common among infants born in tertiary care facilities in India (relative risk [RR] 1.98 [95% CI 1.72–2.19]; **Table S2**) and among infants delivered by caesarean section versus vaginal delivery (RR 1.31 [1.05–1.53]). All neonatal rotavirus infections were asymptomatic.”*

2.20) Line 118 – Prior infection was associated with a reduced likelihood of shedding ORV
Given reference to only seroconversion in line 116, please define “prior infection” are these seropositive infants or seropositive +/- pre-dose 1 shedders? See also above

Author response: As per the comment above, we now use the term “Neonatal infection” rather than “Prior infection” here, thus clarifying that we refer to the defined subgroup above (shedding and/or baseline seropositivity).

2.21) Line 119-120 – “by contrast, pre-vaccination infection did not influence the likelihood of seroconversion”

the authors have two definitions of seroconversion (for >20 IU for seronegative infants and 4-fold titer increase for seropositive) – please clarify the baseline population and what was analysed here.

Author response: As described above, we now use “neonatal infection” rather than “pre-vaccination infection” here, thus clarifying the population being referred to. The original definition of seroconversion (see comment 2.16 for details) holds and we have therefore opted not to redefine it here.

2.22) Line 119-121 – By contrast pre-vaccination infection did not influence the likelihood of seroconversion...and where ORV replication was observed...

again, this sentence is confusing. Separate these two findings if the populations that were being analyzed are not the same. It is not clear from the text what the second part of this sentence is describing (only from the figure) “where ORV replication was observed” – the figure suggests the authors mean post dose 1. Also remove term ‘replication’.

Author response: Thanks for this helpful comment. As suggested, we have (i) separated this into two separate sentences; (ii) clarified that we are referring to shedding after either dose; and (iii) replaced “replication” with “shedding”. We agree that the final wording is much clearer (line 145):

*“By contrast, neonatal infection did not influence the likelihood of seroconversion (RR 1.25 [0.86–1.73]). Where ORV shedding 1 week after either dose was observed among infants with neonatal infection, this significantly boosted post-vaccination RV-IgA concentrations... (**Figure S1B**).”*

2.23) Also – suggest that authors provide figure that is identical to figure 1E, but shows proportion seroconversion (instead of using figure S1B)

Author response: As suggested, we have removed **Figure S1B** as key details are covered in the text and Supplementary figures. We have also moved **Figure 1E** to the supplement (it now replaces **Figure S1B**) so as to simplify **Figure 1**.

However, we have opted not to add a corresponding figure showing seroconversion rates. The key point of **Figure 1E** is to highlight the cumulative effect of neonatal infection and ORV. We have therefore modified the text to reflect this (line 146):

“Where ORV shedding 1 week after either dose was observed among infants with neonatal infection, this significantly boosted post-vaccination RV-IgA concentrations, pointing to a cumulative effect of neonatal infection and vaccination on immunogenicity (Figure S1B). Indeed, while the post-vaccination GMCs of Indian infants lacking neonatal infection were commensurate with those observed in Malawi (6 [5–8] vs 9 [6–12], respectively), the final antibody levels among Indian infants with neonatal infection exceeded those observed in the UK (55 [42–73] vs 27 [17–45], respectively).”

2.24) Figure 1E – define “pre-ORV infection” and “any vaccine shedding”

Author response: We now use the terms “neonatal infection” and “post-ORV shedding, 1 week after either dose” in the figure, thereby harmonising with the wording in the text and in **Figure 1B**. We also provide the following clarifying details in the figure legend (line 573):

“Post-ORV shedding was detected via quantitative PCR targeting the Rotarix NSP2 gene. Neonatal infection was defined as the detection of wild-type rotavirus shedding in week of life 1 or baseline seropositivity (RV-IgA ≥ 20 IU/ml before dose 1).”

Breastfeeding, growth, and sanitation

2.25) Line 131-2 – suggest to help the reader by explaining the significance of the dry season in India on RV circulation

Author response: Thanks for this suggestion. Our primary aim here is to highlight the fact that season might have contributed to the observed trend given that OPV schedule was not randomly allocated (due to the nature of the synchronised tOPV–bOPV switch). Seasonal variation would affect not only rotavirus but also other exposures such as non-polio enteroviruses that may interact with ORV response. To clarify this, we have adjusted the wording as follows (line 157):

“[Seroconversion] was less common in infants who received tOPV-containing than bOPV-only schedules (RRs of 0.56 [0.29–0.95] for tOPV-only and 0.56 [0.29–0.95] for mixed tOPV/bOPV), although the potential contribution of seasonal changes (e.g. in enteropathogen exposure) to this trend cannot be discounted given that OPV schedule was not randomly allocated.”

2.26) Line 133 – suggest to add “were apparent in India”

Author response: Modified as suggested.

2.27) Line 134 – add “in India but not Malawi and the UK” after “first ORV dose”

Author response: We have added “in India” to clarify that the association is specific to this cohort. The lack of significant findings for cofactors in Malawi and the UK is covered by the final sentence of the paragraph (line 166: “Baseline health and demographic variables were not significantly correlated with ORV response in Malawi or the UK...”).

2.28) Line 136 – add “in India”

Author response: Modified as suggested.

2.29) Line 138 – suggest to change “strongly” to “significantly”

Author response: Modified as suggested.

2.30) Line 139 - suggest to add “(Table S5)”

Author response: Modified as suggested, citing **Tables S2 to S4**, where the relevant results are now reported.

Maternal antibodies

2.31) While both are statistically significant, the authors emphasize transplacental over breastmilk antibodies in this section. Suggest to reduce interpretation in results section and/or provide a more balanced interpretation of relative importance of IgG transplacental transfer vs breastmilk IgA.

Author response: This point is well taken. We have shortened the paragraph on serum antibodies, deferring interpretation to the discussion section. For example, as suggested in the comments that follow, we have shortened or cut sentences in the paragraph on serum RV-IgA (now starting on line 178) and expanded the subsequent paragraph on breastmilk RV-IgA (line 195). Specific changes are detailed in our responses below. We feel that the current wording offers a more balanced presentation of the results, and are grateful to the reviewer for proposing these modifications.

In addition to the changes detailed below, we have added a panel to **Figure 2B** comparing maternal breastmilk/serum RV-IgA ratios across the three cohorts, emphasising the fact that breastmilk concentration in Indian mothers were lower than would be expected based on their serum IgA levels. These results are described as follows:

Line 175: "Reflecting the relative deficit in breastmilk RV-IgA levels in India, maternal breastmilk/serum RV-IgA ratios were significantly lower in this cohort than both Malawi and the UK (Figure 2B)."

Line 335: "Nutritional stress has also been linked with lower secretory IgA levels in breastmilk^{23,24}. Consistent with this, we observed maternal breastmilk/serum RV-IgA ratios to be lowest in India and highest in the UK, with intermediate levels in Malawi."

2.32) Line 152, remove "this may reflect a lack of statistical power" and move to discussion. Please also include sample size for all figures

Author response: Modified as suggested.

2.33) Line 154-5 - "Serum IgA is not transmitted efficiently across the placenta and is therefore unlikely to directly influence ORV". What about the correlation between serum IgA and breastmilk IgA? Breastmilk IgA may "directly influence ORV". Suggest to modify sentence to be more specific and/or move to discussion.

Author response: As suggested, we have deleted this sentence and defer the interpretation of these correlations to the discussion.

2.34) Figure 2B - use uniform terminology for previously infected cohort (neo+/- vs infected in other figures)

Author response: As suggested, we now refer to neonatal rather than pre-vaccination infection in the figure legend, thus harmonising with the terminology used above. Specifically, the terms neo+ and neo- are defined as follows in the **Figure 2** legend (line 503):

"neo+, infected with rotavirus neonatally (as defined by detection of rotavirus shedding in week of life 1 or baseline seropositivity); neo-, uninfected with rotavirus neonatally"

2.35) Line 157-159 - "maternal serum RV-IgA levels were negatively correlated with ORV shedding in both India and Malawi (Figure 2C and Table S3)".

Figure 2C shows a significant negative correlation between maternal IgA and shedding in India - primarily driven by the neo+ group, however Table S3 shows no significant correlation for maternal serum IgA and shedding in Malawi after dose 1 or after either dose. Please support conclusion with significant results or remove.

Author response: Thank you for this valuable comment. We have modified the wording to: (i) reflect the fact that the results in Malawi were not statistically significant; (ii) cite specific data for dose 1 shedding; and (iii) highlight the stronger effect size among Indian infants with neonatal infection. The final wording is as follows (line 185):

*“Second, maternal serum RV-IgA levels were negatively correlated with ORV shedding after dose 1 in India (RR 0.68 [0.53–0.85]; **Figure 2C**), particularly among infants with neonatal infection (RR 0.50 [0.31–0.75]), and a similar trend was evident in Malawi (RR 0.85 [0.67–1.00]; **Table S3**).”*

Despite that lack of statistical significance in Malawi, we believe that the trend merits comment given its similarity with the data from India.

2.36) Line 160 - It appears that maternal antibodies inhibited both neonatal antibody anti-RV responses and post-vaccination anti-ORV responses, but not neonatal RV shedding in the Indian infants. How do the authors interpret this?

Author response: We now address this in the Discussion section as follows (line 369):

“Neonatal infection with the G10P[11] strain was not sensitive to the inhibitory effect of maternal antibody levels (a trait also reported for the G3P[6] oral human neonatal rotavirus vaccine candidate RV3-BB²⁴), potentially reflecting serotype-specific adaptation to the newborn gut²⁵ alongside reduced exposure to RV-IgA (and other antiviral compounds) if infection occurs prior to breastfeeding.”

2.37) Paragraph 166, authors suggest that breastmilk IgA may be less inhibitory than transplacental antibodies, however maternal serum IgA and breastmilk IgA both correlate negatively with shedding whereas maternal IgG and cord blood IgG do not. This data suggest that there may be divergent mechanisms driving correlations between breastfeeding and antibody and shedding responses. Suggest to modify sentence on line 169 or move to discussion.

Author response: As suggested, we have amended this paragraph to focus on results, deferring interpretation to the Discussion section. The sentence formerly on line 169 (“However, as noted above...”) has been removed. The paragraph now reads as follows (line 195):

*“Disentangling the relative influence of breastmilk versus transplacental antibodies is challenging given the correlation between the two (**Figure 2D**). As observed for maternal serum RV-IgA, breastmilk RV-IgA was not significantly correlated with seroconversion in any cohort (**Table S4**), but was negatively correlated with infant post-vaccination RV-IgA levels in both India and Malawi (**Figure 2C**). Breastmilk RV-IgA was negatively correlated with ORV shedding after dose 1 in India (RR 0.83 [0.69–0.98]; **Table S3**) but not Malawi (RR 0.91 [0.58–1.23]).”*

Inflammatory biomarkers

2.38) Why were the same inflammatory biomarkers not measured across all countries?

Author response: MPO and α 1AT were assayed in all countries. α 1AG were omitted for the UK owing the limited pre-vaccination serum volumes available for this cohort. We have clarified this in the Methods as follows (line 783): “ *α 1AG assays were not performed for the UK so that the limited pre-vaccination sample volumes for this population could be prioritised for RV-IgA assays.*” A similar clarifying sentence has also been added to the legend of **Figure S2** (line 583): “ *α 1AG assays were not performed for the UK owing to the limited sample volumes available for this population.*”

Geographic differences in microbiota development

2.39) Line 180-1 – “We obtained high quality faecal microbiota profiles in 2,086 samples”

Please provide total number of samples sequenced (2,138?)

Author response: We have clarified these details as follows (line 210): “*We sequenced 2,137 separate faecal samples from the study population, of which 2,086 yielded high-quality microbiota profiles ($\geq 25,000$ sequences after quality filtering; $142,880 \pm 136,113$ [mean \pm s.d.] sequences per sample).*”

2.40) Line 188-191 Why do authors only name the genera enriched in India and UK but not in Malawi (suggest to either remove from text or add major genera enriched in Malawi)

Author response: This was a pragmatic decision given the large number of enriched genera in Malawi. To address this, we now clarify the number of enriched genera and list a subset, as follows (line 220):

*“Based on longitudinal mixed-effects models, 13 genera (including *Prevotella*, *Sutterella*, *Corynebacterium*, and *Acinetobacter*) were enriched across infancy in Malawi compared with both India and the UK (Figure S5).”*

2.41) Figure 3C report significance by country

Author response: To address this, we have amended **Figure 3C** (now **Figure 3B**) to report the R^2 and significance level of PERMANOVA tests for country at each sampling time point. This is clarified in the text as follows (line 218):

“Inter-individual differences accounted for 58% of variation in microbiota composition based on permutational multivariate analysis (PERMANOVA; $p < 0.001$, 999 permutations), while country accounted for 6–10% of variation depending on age (Figure 3B).”

2.42) Sentence 198-199 “Indeed, simple cross-sectional comparisons of taxon prevalence were adept at selecting differential microbiota colonization patterns...”

what extra information does this provide beyond what is additionally presented in figure S5A and B? Suggest to streamline and remove.

Author response: We believe that these cross-sectional analyses offer a valuable adjunct to the longitudinal models as they have the potential to capture discrepancies that are significant at individual timepoints but not significant in longitudinal analyses. We have therefore retained the analyses and slightly modified them (as suggested in comment 2.48) to combine differential prevalence (Fisher’s test) and differential abundance (Aldex2). Effect sizes and p values for both metrics are summarised in **Figure S5C** and **Table S5**, and referred to in the text as follows (line 225): “Numerous other discriminant taxa were identified during cross-sectional analyses of genus prevalence and abundance at each sampling timepoint (**Figure S5C** and **Table S6**).”

Microbiota composition versus ORV response

2.43) Line 204-5 “Interestingly, stratified analyses”
add “in India”

Author response: Modified as suggested.

2.44) Line 201 – the authors describe an interesting negative correlation between microbiota alpha diversity and ORV immunogenicity that appears to be the most pronounced in the non-rotavirus exposed group in India and the time of first ORV vaccination in Malawi. It is surprising, given the significance in alpha diversity, that there are no corresponding differences in taxonomic composition/beta diversity according to ORV response. The authors have used Random Forest models for their cross-sectional analysis and also present a longitudinal analysis in figure 4E. Please provide a rationale for why these methodologies were used.

Author response: The cross-sectional and longitudinal approaches provide complementary insights into the microbiota. Whereas the former might highlight differences specific to a particular timepoint (e.g. differences in taxonomic composition at the time of the first dose of ORV), the latter may capture discrepancies that accrue across early life but are perhaps not be significant at any individual timepoint. We have added the following to the methods to clarify this difference in rationale (line 923): *“To complement the cross-sectional analyses at each sampling timepoint, we took advantage of the longitudinal study design to explore variation in microbiota diversity and genus abundance across early infancy.”* The description of the longitudinal models follow on from this sentence.

Given the differences in alpha diversity, the reviewer requests a more detailed analysis of beta diversity. The reviewer requests the following:

2.45) - Analysis of differences in taxonomic composition according to ORV response at the sequence variant level (github analysis suggests both genus and sequence variant level analyses were done)

Author response: As suggested, we now present Random Forest model accuracy and differential taxonomic abundance statistics at both genus and sequence variant level for seroconversion (**Figure 4C**), post-vaccination RV-IgA (**Figure S7C**) and dose-1 shedding (**Figure S9C**). After FDR adjustment, we observed no differentially abundant taxa in relation to any of these three ORV endpoints. These additions to the analyses are clarified in the text as follows (line 260):

*“Cross-sectional Random Forests models based on genus or ribosomal sequence variant (RSV) abundances failed to accurately predict seroconversion (**Figures 4C**), post-vaccination RV-IgA (**Figure S7C**), or ORV shedding after dose 1 (**Figure S9C**). After FDR correction, we did not observe significant differences in the prevalence (based on Fisher’s exact test) or abundance (based on Aldex2) of individual genera or RSVs according to ORV outcome.*

2.46) - Given the difference in alpha diversity by seroconversion for the neo- population, the reviewer requests an analysis of difference in alpha and beta microbiome composition between neo + and neo – infants in India. Additionally, it is striking that, only for the infants in the neo- group, infants are significantly more likely to seroconvert if they have had a caesarian section (particularly as infants with caesarian section are known to have slower colonization rates). Please also compare the alpha and beta microbiota diversity of infants with and without a cesarian section in this group to evaluate if delivery mode may be impacting/confounding alpha diversity/microbiome maturation and seroconversion in this subgroup.

Author response: Thank you for this valuable suggestion. We agree that a more in-depth analysis of alpha and beta diversity in relation to neonatal rotavirus infection and other cofactors would be valuable, and have added a new figure accordingly (**Figure S6A**). We have added the following paragraph describing these results (line 230):

*“To explore cofactors associated with the developing microbiota in each cohort, we performed an exploratory analysis of alpha and beta diversity among samples collected at the time of the first ORV dose (**Figure S6A**). Based on PERMANOVA of genus-level unweighted Bray–Curtis distances, microbiota composition in India was significantly correlated with a1AT (R² 3.6%), breastfeeding status (R² 2.0%), age at vaccine delivery (R² 1.3%), and delivery mode (R² 1.1%), though only breastfeeding status was significantly correlated with microbiota diversity (mean [s.d., n] Shannon index of 1.2 [0.4, 248] in exclusively breastfed infants vs 1.5 [0.4, 41] in infants with partial or no breastfeeding; FDR p <0.001). Neonatal rotavirus infection was not significantly correlated with microbiota composition or diversity (**Figure S6B**). In Malawi, microbiota composition was associated with maternal RV-IgA level (R² 4.8%), a1AT (R² 4.8%), HIV exposure (R² 4.6%), and maternal age (R² 4.1%). a1AT and maternal RV-IgA were positively correlated with microbiota diversity. No covariates were significantly associated with alpha or beta diversity in the UK after FDR correction, although this fewer variables were captured in this cohort.”*

As noted above, neonatal rotavirus infection status was not associated with either beta diversity or alpha diversity, and we present a full longitudinal analysis of Shannon index in **Figure S6B**.

To address the potential confounding effect of C-section delivery, we have added a sensitivity analysis of alpha diversity in which these infants are excluded. We have done a similar sensitivity analysis restricted to exclusively breastfed infants. Both revealed the main effect among Indian infants lacking neonatal infection to be unchanged. These sensitivity analyses are reported in **Figure S8** and in the text as follows (line 249):

“These associations were robust in sensitivity analyses restricted to exclusively breastfed infants or those born by vaginal delivery (p values of 0.008 and 0.004 for respective longitudinal models of Shannon index; Figure S8).”

2.47) - Alpha diversity is lower in UK than Malawi but similar between UK and India. Is UK alpha diversity lower than Indian neo- alpha diversity? In other words, is there a consistent trend between low alpha diversity and increased seroconversion across countries and within countries when the neo+ group is removed?

Author response: As noted above, we have added **Figure S6B** which highlights that alpha diversity did not differ significantly between neo+ and neo- infants in India. Since there is no evidence that neonatal rotavirus infection status significantly alters alpha diversity, we have not repeated the full geographic comparisons.

2.48) - Please perform a complementary differential abundance analysis to the cross-sectional random forest models, such as DESeq or equivalent that is tailored to microbial genomic data, to orthogonally evaluate differential abundance by seroconversion and shedding per time point, particularly for neo-group in India and at time of first vaccination for infants in Malawi

Author response: As suggested, we have added differential prevalence (using Fisher’s exact test) and abundance (using Aldex2) comparisons according to ORV outcome to the cross-sectional analyses. These results are tabulated below the relevant Random Forest models (**Figure 4C**, **Figure S7C** and **Figure S9C**). Notably, no significant associations were observed after FDR correction.

2.49) - Please evaluate if there are significant differences in microbiome composition according to demographic outcomes that correlate with ORV immunity and could be confounding microbiome analyses– including delivery in a tertiary care facility, delivery by cesarian section, exclusive breast feeding, height for age

Author response: As suggested, we now report a full analysis of alpha and beta diversity in relation to various cofactors (**Figure S6**). See comment 2.46 for further details.

2.50) - Figure S7 – figure E, what is the asterix referring to? figure D appears incorrectly labeled in text (should be F).

Author response: To simplify the presentation of data, we have now removed this figure and present the longitudinal abundance comparisons in **Table S7**.

2.51) - Line 218 – what do the authors mean with “microbial exposure” and how is this supported by their findings? Suggest to remove. Did the authors consider calculating a Maturation index for microbiome development to support this notion of colonization?

Author response: Thank you for highlighting this ambiguity. We have amended the wording to clarify that we are referring to ‘microbiota diversity’, as presented in **Figure 4A**. A maturation index would be an interesting avenue, although these models have typically been validated across a wide age range (e.g. 0–2 years in Subramanian et al, Nature 2014) whereas our study focuses on the first 2–3 months of life. With this in mind, we deemed such analyses to be beyond the scope of this already complex analysis.

Multivariate analysis

2.52) Please indicate in figure 5 and in the title of its legend that this analysis is only of Indian infants.

Author response: As suggested, we have added a note at the top of **Figure 5A** specifying that the analysis refers to the 249 Indian infants with complete data. Likewise, we have amended the figure legend as follows (line 544): *“Integrated Analysis for Prediction of Oral Rotavirus Vaccine Response in Indian Infants.”*

2.53) Authors include pre-vaccination microbiota composition in the multivariate analysis. Please describe in more detail what criteria were used for the 126 genus abundances.

Author response: We have harmonised all Random Forests models to include taxa detectable in $\geq 5\%$ of samples of one or more of the subgroups being compared. This results in 55 genera being included in the integrated analysis described in this section. We have clarified this in the Methods section as follows (line 954): “... and genus relative *abundances at the time of the first ORV dose (55 variables, encompassing genera present at $>5\%$ of samples).*”

2.54) Line 230 - the authors should better describe the results presented in figure 5 in the text – they now describe them without contextualizing them to the cohorts (all India/neo +/neo-) analyzed. Suggest to address: Pre-vaccination rotavirus infection is the most important predictor of IgA concentration for the Indian population as a whole and maternal IgG concentration in the exposed cohort predicts IgA concentration. However, for non-exposed infants, the effect sizes of variables is considerably diminished and multiple microbiota and antibody variables correlated with IgA concentration.

Author response: Thanks for this valuable critique. We have added the following sentence to contextualise the cohorts included in these analyses (line 275): “*These analyses focused on Indian infants given the larger size of this cohort (n = 249 with complete data). Moreover, we included analyses stratified by neonatal rotavirus infection status given its modifying effect on the associations reported above.*”

As suggested, we have also expanded the description of these results by adding the following (line 281):

“Prediction of post-vaccination RV-IgA was markedly reduced in the stratified analyses (R^2 of 16.1% [9.6–22.1%] and 3.7% [0.3–5.1%] in infants with and without neonatal infection, respectively). Maternal RV-IgG was the most important predictor of post-vaccination RV-IgA among infants with neonatal infection, whereas multiple maternal antibody and microbiota variables were among the top-ranking features for infants without neonatal infection (Figure 5B).”

2.55) A shortcoming of these regressions is the repeated use of post-vaccination IgA concentration as outcome– which this paper itself shows is an imperfect correlate of protection. Were the same outcomes found when correlating with rotavirus vaccine shedding?

Author response: Full univariate and multivariate results for seroconversion and dose 1 shedding are provided in Table S3 and Table S2, respectively. We have added the following text to summarise these results (line 288):

“Consistent with the Random Forests outputs, few significant predictors were identified for multivariate models of seroconversion among Indian infants (Table S3). ORV shedding after dose 1 in this cohort was negatively correlated with neonatal rotavirus infection (RR 0.51 [0.30–0.80]) and maternal RV-IgA (RR 0.68 [0.53–0.85]), and was higher in the sequentially recruited cohort of IPV recipients (RR 2.18 [1.41–3.08]; see Table S2 for full results).”

Discussion

2.56) The authors have analyzed and collated a dense and impressive degree of geographic, demographic, immune, and microbiome risk factors for rotavirus vaccine performance. This makes the discussion especially important in leading the reader through the significant and most important non-significant findings. However, in the discussion, the authors are summarizing their findings as according to their pre-existing sections and do not prioritize their findings or offer the reader an overarching interpretation of their work. Foremost among the open questions is if the authors believe the risk factors they evaluated and their study findings (singularly or in combination) adequately explain the diminished protection rotavirus vaccines afford infants in LMIC. Did they meet their study aim? And secondly, how their findings may have shifted understanding of key questions around rotavirus vaccine performance – eg. that enteropathy/low diverse microbiota may not be driving poor performance; the potential importance of early (neonatal rota) viral exposures; the

complexity of maternal antibody responses and vaccine performance. And while they list future directions, what work deserves priority in the future?

Author response: This is a valuable critique, and we recognise the need for an over-arching synthesis of the findings. With this in mind, we have added the following paragraph towards the end of the Discussion, highlighting our overall take on the study objectives as well as the areas of future research that ought to be prioritised (line 403):

“Despite the multiple demographic, immunological, and microbiota variables considered, our overall ability to account for discrepancies in ORV response within and between cohorts was limited. Alternative risk factor for impaired ORV outcome must therefore be considered. In particular, we would advocate further efforts to profile early-life host–microbe interactions in LMICs, including high-throughput measurements of breastmilk (e.g. metabolome, microbiome, and antibody functionality) and faecal samples (e.g. virome and metabolome). This will help to determine the extent to which metabolic and microbial exposures combine to inhibit ORV response over the early weeks of life. The epigenome merits further consideration given that BCG-induced epigenetic remodelling of monocytes has been linked with protection against heterologous viruses³⁵. It is also possible that we considered the correct mechanisms but failed to capture them with sufficient granularity using the methods and samples available. Alternative EED markers may be required to accurately characterise the onset of this condition in early life. Moreover, given the potential for overall microbial load to vary substantially from infant to infant, absolute as opposed to relative quantitation of microbial abundances may be necessary to capture the interplay between the developing bacterial microbiota and ORV outcome³⁶.”

2.57) Line 262-3 We advocate the further use of vaccine virus replication as an adjunctive measure of ORV response in future trials

While the reviewer agrees that rotavirus vaccine shedding may be a better correlate of protection than post-vaccination IgA, this work has not evaluated vaccine shedding in relation to clinical protection from severe rotavirus gastroenteritis, and therefore it is unclear on what basis the authors are advocating this standpoint. Additionally, there are numerous technical difficulties in using shedding as an outcome which also deserve to be addressed – if serial samples are not obtained, which day post-vaccination should be used? What is the likelihood that shedding can be missed?

Author response: Thank you for this comment. Our intention was to highlight the potential value of shedding as a way of probing divergent vaccine outcomes rather than as a validated correlate of protection. We have amended the wording accordingly, and have also commented on the potential value of measuring multiple timepoints where feasible (line 319):

“Although ORV shedding has not been validated as a correlate of protection, our study highlights the potential value of this outcome in capturing divergent ORV responses. We advocate the further use of vaccine virus shedding in the week after delivery (at multiple timepoints where possible) as an adjunctive measure of ORV response in future trials.”

2.58) Lines 277-8 – “we observed a reduction in both shedding and immunogenicity, suggesting that the effect of maternal antibodies is mediated in part through a reduction in ORV replication efficiency”

the reviewer disagrees that authors have demonstrated that maternal antibodies had a clear effect on shedding alongside immunogenicity. The authors demonstrate a significant negative correlation between maternal antibodies and post-ORV IgA. However, the correlation with shedding is only for maternal IgA and only in the neo + group in India – not in any other cohort or subgroup. This makes extrapolation from maternal antibodies to shedding purely speculative. They have also not demonstrated an effect on replication as they have not measured replication. That the effect is only seen in the neo + group suggests instead that a negative correlation of maternal antibodies on shedding in exposed infants may be mediated instead through an effect on infant IgA/IgG pre-ORV. Please address and modify rest of paragraph accordingly.

Author response: Although the reviewer is right to point out that the trend is strongest in neo+ Indian infants, the negative correlation is significant for the Indian population as a whole (the primary comparison group), while the same trend is apparent for breastmilk IgA in neo- Indian infants (RR 0.86 [0.69–1.02] for dose 1 shedding) and for maternal serum RV-

IgA in Malawian infants (RR 0.85 [0.67–1.00]). We therefore feel that the finding merits comment, especially given the corollary at the end of the paragraph (“these mechanisms alone are clearly insufficient to prevent a robust ORV response”).

However, we acknowledge that the conclusion should not be overstated, and therefore have amended “*is mediated in part*” to “*may be mediated in part*”. Moreover, as detailed below (comments 2.59 and 2.60) we have cut several sentences expanding on potential mechanistic context in this paragraph. We feel that the final wording is more balanced, highlighting these interesting findings without overinterpreting them, and thank the reviewer for their input here.

2.59) Line 281 – relevance of passively acquired IgG antibodies and lentiviruses is unclear, suggest to remove

Author response: Modified as suggested.

2.60) Line 283 – the authors have not shown that there is a leaky mucosal barrier or early-life enteropathogen exposure in their populations – rather their results suggest the opposite, that EED biomarkers do not differ significantly by ORV response. Please address and suggest to remove speculation not in line with findings.

Author response: As suggested, we have removed this sentence to avoid potential confusion.

2.61) Line 302 – suggest to change language “...oral vaccine failure and/or that the condition...(quite possibly both)”

Author response: We believe that the reviewer is requesting the use of ‘and/or’ in place of ‘(quite possibly both)’ and have modified accordingly.

2.62) Line 303 – reviewer suggests that authors also address microbiota differences within country (e.g. was the neo + and neo – group significantly different in alpha and beta diversity)

Author response: As noted above (comment 2.46), we have added **Figure S6** and an additional results paragraph (line 230) reporting on cofactors associated with microbiota composition in each cohort. To ensure that the Discussion remains focused on the primary outcomes (differences by geography and ORV response), we have opted not to comment further on these associations in this section, though we refer to the sensitivity analyses restricted to breastfed/vaginally delivered infants (see comment 2.64 below).

2.63) Line 310 please address that alpha diversity results found within countries showing that higher alpha diversity correlates with lower immunogenicity are not found between countries (where one would expect lower diversity in UK compared to both India/Malawi)

Author response: We have highlighted this important caveat by adding the following (line 389): “*Given that infant microbiota diversity was comparable in India and the UK, this factor alone is clearly insufficient to account for broad geographic trends in ORV response.*”

2.64) Line 315 – address possible correlation between cesarian section and microbiome diversity in India.

Author response: As shown in **Figure S6**, delivery mode was correlated with composition but not microbiota diversity. Moreover, a sensitivity analysis restricted to Indian infants born by vaginal delivery did not affect the observed association between diversity and seroconversion. We have addressed this in the Discussion by adding the following sentence (line 387): “*In India, the correlation was strongest in infants lacking neonatal rotavirus infection, and was robust in sensitivity analyses restricted to exclusively breastfed or vaginally delivered infants.*”

2.65) Line 315 –For example, microbiota richness may act as a proxy for early life exposures that shape ORV outcome, such as non-polio enteroviruses
the reviewer does not completely follow the authors’ rationale here – low diversity correlates with increased seroconversion. The reviewer would assume viral infection/non-polio enterovirus exposure would result in reduced microbiota diversity rather than increased diversity (as has been

shown with numerous enteric viral and bacterial diarrheal diseases). Then NPEV would be positively correlated with seroconversion. Whereas in the literature NPEV and OPV are negatively correlated with seroconversion (Taniuchi, Vaccine 2016).

Author response: Thank you for pointing out this ambiguity. We agree that the logic was convoluted, so have cut these two sentences to streamline this section of the Discussion. We now mention the potential significance of non-bacterial components of the microbiome as follows (line 392):

"It is plausible that early-life exposure to greater microbial diversity (including members of the virome and eukaryome not measured here) may foster a state of hyporesponsiveness at the mucosal epithelium that impairs oral vaccine outcome in LMICs."

2.66) Line 317 – The fact that ORV immunogenicity was impaired among Indian infants in households without access to treated water would support this notion

However, other studies have shown that infants without access to treated water and poor living conditions generally have lower microbial diversity and microbiome “immaturity” (Subramanian, Nature 2014), yet the current study has found that infants with lower diversity have higher seroconversion. Please address.

Author response: As noted above (comment 2.65), this sentence has now been cut.

2.67) Line 323-4 – a clear understanding of the signature of ‘healthy’ microbiota development in different settings is lacking

Please revise to address growing body of literature on microbiome diversity and malnutrition among young children in low-income settings such as Malawi and Bangladesh (e.g. Smith M, Science 2013; Chen R, NEJM, 2020; Gehrig J, Science 2020)

Author response: We agree that this body of work merits inclusion, although it is worth noting that the studies focus predominantly on associations in later infancy (e.g. 6–36 months in the study by Gehrig et al), whereas studies focusing specifically on the first 3 months of life are scarce. To address this, we have cut the line referring to healthy microbiota development and added the following text, which includes the suggested citations (line 396):

"Our findings in Indian infants also support the notion that the neonatal period represents a key window of microbiome development²⁹. Indeed, associations between microbiota diversity and ORV response in this cohort were apparent from the first week of life onwards. While a growing body of literature has emphasised the importance of microbiota composition in relation to health outcomes such as malnutrition among children in LMICs^{30,31}, studies focusing on the first days and weeks of life are presently lacking, and represent a crucial avenue of future research."

2.68) Line 337 – this study advances our understanding of the potential mechanisms influencing ORV response in several ways

The reviewer agrees with this statement and this work is a very comprehensive evaluation of possible known risk factors for impaired rotavirus vaccine performance in LMIC. However, the work has not yet uncovered clear answers about why rotavirus vaccines are insufficiently immunogenic in these populations. The reviewer requests that the authors speculate about whether risk factors are missing or if they believe their work demonstrates that a combination of factors is driving the observed differences in vaccine efficacy between high and LMIC or if they used the wrong correlates of vaccine protection.

Author response: As detailed in comment 2.56, we have added a paragraph specifically addressing alternative mechanisms that we deem to a priority for future research.

2.69) Line 341 – the reviewer disagrees with the authors conclusion that they have shown that maternal antibodies reduce ORV replication. See comments to lines 277-8 above, in addition they have not evaluated replication, only looked at the presence or absence of ORV shedding. Suggest to remove or provide evidence to support this claim.

Author response: As suggested, we have replaced “replication” with “shedding” and toned down the statement by replacing “is mediated” with “may be mediated”. See comment 2.58 for further details.

2.70) Line 341-44 – neonatal rotavirus exposure in India is one of the most striking observations in this paper, with by far the largest reported effect size (Figure 5B). Given this was one of the most significant findings in the manuscript, this deserves far more weight in the discussions. Please address.

Author response: We thank the reviewer for this suggestion and have added the following paragraph to the Discussion to address this (line 365):

“A key finding of translational relevance is the highly immunogenic nature of neonatal rotavirus infection in India. Neonatal infection was detected in almost half of the infants in this cohort, and these infants exhibited significantly higher final RV-IgA concentrations than non-shedders. Machine-learning models corroborated this by selecting neonatal rotavirus infection status as the most important determinant of final RV-IgA concentration out of 85 input variables. Neonatal infection with the G10P[11] strain was not sensitive to the inhibitory effect of maternal antibody levels (a trait also reported for the G3P[6] oral human neonatal rotavirus vaccine candidate RV3-BB²⁷), potentially reflecting serotype-specific adaptation to the newborn gut²⁸ alongside reduced exposure to RV-IgA and other antiviral compounds if infection occurs prior to breastfeeding. Trials exploring the shedding and immunogenicity of Rotarix (a G1P[8] strain) when administered neonatally would help distinguish these possibilities. In Indonesia, RV3-BB efficacy against severe rotavirus gastroenteritis was 75% when administered according to a neonatal schedule (weeks 1, 8 and 14) and 51% when administered on a later schedule (weeks 8, 14 and 18), highlighting the efficacy of schedules that include ORV at birth²⁹.”

2.71) Line 345 – the reviewer is not sure what the authors mean to suggest with ‘high microbial exposure’. Do the authors mean enteropathogen exposure? More rapid colonization? Higher alpha diversity? Please specify and address and support the corresponding hypothesis in discussion prior to using this as a concluding statement.

Author response: Thank you for highlighting this ambiguity. We were referring to the microbiota diversity results, and have amended the wording to be more explicit (line 438): “... high microbiota diversity in early life may contribute to the impaired efficacy of this vaccine in LMICs.”

Methods

2.72) Study design

Please describe when shedding samples were collected in relation to vaccination (days) and over what range of time collection were considered acceptable for inclusion.

Author response: We have clarified these details as follows (line 706):

“Infants provided two blood samples (weeks of life 6 and 14 in India and Malawi; weeks of life 8 and 16 in the UK) and four stool samples over the course of the study (Figure 1A), including samples at the time of each ORV dose (weeks of life 6 and 10 in Malawi and India; weeks of life 8 and 12 in the UK). For post-vaccination rotavirus shedding assays (weeks of life 7 and 11 in Malawi and India; weeks of life 9 and 13 in the UK), stool samples provided within ±1 days of the target were considered eligible for inclusion. For all other timepoints, serum and stool samples provided within ±3 days of the target were considered eligible.”

2.73) Line 550 Why was cord blood and maternal serum collected but no maternal serum IgG, cord blood IgG reported for Malawi and UK? Please explain or include this data

Author response: We thank the reviewer for pointing out this deviation from the published protocol, which we agree needs to be more specifically addressed. Although samples from the UK and Malawi were assayed for RV-IgG, a technical issue with the ELISA plates resulted

meant that the data did not pass the QC requirements for inclusion in the analysis. We have addressed this in the Methods section as follows (line 739): *“However, a technical issue with the assay precluded the inclusion of RV-IgG data from the UK and Malawi.”*

2.74) Line 601 What was the range of the shedding Ct values used for correlation analyses.

Author response: We have clarified the range of shedding Ct values in the legend to Figure 2 as follows (line 500): *“Shedding after week of life 1 was determined based on the group A rotavirus VP6 gene assay (Ct range 23.5–35.0) while shedding after dose 1 was based on the Rotarix-specific NSP2 gene assay (Ct range 20.7–40.0).”*

2.75) Line 640 Why were MPO and alpha1AT not performed for UK infants?

Author response: Addressed in our response to comment 2.38.

2.76) *Statistical analysis*

Provide a rationale for the study sample size including a rationale for why the sample size is so different across the three sites. Please also address this in limitations of the study in the discussion. In the published study protocol (Sindu BMJ, 2017), the sample size was estimated to require 150 mother infant pairs in India and Malawi and 55 in the UK for infant seroconversion outcomes. Why was sample size so much lower in Malawi and what impact may this have had on study results? Also discuss the lower seroconversion than expected in the UK and how this relates to the power calculations that were made for the UK which assumed > 95% seroconversion.

Author response: Thank you for this comment. We agree that these points merit additional clarification. We have added a new section to the Methods (line 957), clarifying the sample size calculations and deviations in the final analysis:

“As reported in the published protocol³⁷, we calculated that a sample size of 150 infants in India and Malawi would provide 80% power to detect a two-fold higher mean concentration of RV-IgG in infants who fail to seroconvert compared with those who seroconvert (assuming 40% seroconversion in each), while a sample size of 50 in the UK would provide 79% power to detect significant differences in RV-IgA by seroconversion status across the study sites (assuming 95% seroconversion in the UK). These sample sizes would also provide 95% power to detect significant differences in Shannon index according to seroconversion status in India and Malawi. The sample size in India (n = 307) exceeded these targets owing to the high recruitment rates in this cohort and the decision to merge the IPV and OPV arms in the final analysis. On the other hand, owing to challenges in recruitment and sample collection over the course of the study period, the final sample size in Malawi, (n = 119) fell short in these estimates. Since RV-IgG data were not available for the UK and Malawi, our final analyses of maternal antibodies across cohorts focused on RV-IgA.”

As per the power calculations, we now present a stratified analysis of post-vaccination RV-IgA among infants who successfully seroconverted (line 129): *“Among infants who seroconverted, post-vaccination RV-IgA levels did not differ significantly among cohorts (GMCs of 93 [73–118], 122 [80–187], and 105 [71–155] in India, Malawi, and the UK, respectively; Tukey’s post-hoc p values >0.05).”*

As suggested by the reviewer, we have also addressed these limitations directly in the Discussion section by adding the following text (line 420):

“Second, challenges in sample recruitment in Malawi meant that the final study population (n = 119) fell short of the target sample size (n = 150). This was exacerbated by incomplete sample availability for each recruited infant, such that several comparisons in this cohort may have been underpowered. Relatedly, comparisons of post-vaccination RV-IgA across cohorts may have been underpowered given the lower-than-expected seroconversion rates in the UK.”

2.77) Line 741-2 antibody concentration is assumed to be linear, why was Spearman used over Pearson’s?

Author response: Pearson's r was used for the primary correlation analyses between maternal and infant RV-IgA. Spearman's test was only used for the correlation network presented in **Figure 2C** owing to the inclusion of shedding data (1/Ct). The skewed nature of the shedding data makes a non-parametric test more appropriate. We have modified the wording as follows to clarify that the non-parametric test applies only to the correlation network (line 901): *"In Indian mother–infant dyads, a correlation network was also calculated between all measured antibody concentrations (RV-IgG and RV-IgA) and rotavirus shedding quantities (reciprocal of qPCR Ct value) using Spearman's rank correlation coefficients."*

2.78) 741-2 authors used 1/Ct value for shedding – however what was considered a negative Ct value and how was quantification validated? Shedding was evaluated twice, how did the authors use multiple shedding results in their correlation analyses?

Author response: As noted above (comment 2.74), we now clarify the Ct value cut-offs for the NSP2 and VP6 assays on line 767: *"applying maximum Ct thresholds of 40 and 35, respectively"*.

We thank the reviewer for pointing out the ambiguity regarding the use of multiple shedding assays. We can confirm that only one value was used in the correlation network – VP6 Ct for week of life 1 and NSP2 for post-vaccination shedding. We have addressed this in the Figure 2 legend as follows (line 500): *"Shedding after week of life 1 was determined based on the group A rotavirus VP6 gene assay (Ct range 23.5–35.0) while shedding after dose 1 was based on the Rotarix-specific NSP2 gene assay (Ct range 20.7–40.0)."*

Other modifications:

We describe the number of infants enrolled in each cohort alongside the number who met the primary endpoint (line 712): *"Overall, we enrolled 664 mother–infant dyads (395 in India, 187 in Malawi, and 82 in the UK). The primary endpoint (measurement of seroconversion or dose 1 shedding) was reached by 484 dyads (307 in India, 119 in Malawi, and 60 in the UK)."*

We highlight the link between exclusive breastfeeding and post-vaccination RV-IgA in India during the Discussion (line 353): *"Moreover, exclusive breastfeeding was positively correlated with post-vaccination RV-IgA in Indian infants, suggesting that any inhibitory effect of RV-IgA in milk are offset by the potential benefits of breastfeeding for ORV, such as the buffering of gastric acid."*

We discuss the observed link between stunting and ORV immunogenicity as follows (line 362): *"However, we observed a negative correlation between post-vaccination RV-IgA and height-for-age Z score in Indian infants, corroborating recent data from Zimbabwe linking growth deficits with impaired ORV response¹⁴."*

We detail the methods for the multivariate regression models as follows (line 896): *"Following these univariate analyses, variables with p values of <0.05 were explored in multivariate models for each ORV outcome. Where multiple correlated variables were eligible for inclusion in multivariate models, we minimised multicollinearity by prioritising variables measured closer to the first dose of ORV (e.g. Shannon index at 6 weeks of life vs 4 weeks of life), maternal antibodies in serum versus breastmilk, and maternal serum RV-IgG versus maternal RV-IgA or pre-vaccination infant RV-IgG."*

Reviewers' Comments:

Reviewer #1:

Remarks to the Author:

The authors have revised this manuscript extensively in response to the reviewers' comments. My concern were principally with the presentation of data in figures and supplement. I t=consider that these comments have been fully addressed in the revised manuscript.

Reviewer #2:

Remarks to the Author:

Parker and Bronowski et al have significantly revised the manuscript and adequately addressed reviewer comments.

This is a rich and complex data set derived from a thoughtfully-designed prospective longitudinal study that has been well analyzed and will have importance to the rotavirus and vaccinology fields and global health.

Some minor comments

Line 42/43 in abstract, In both India and Malawi, pre-vaccination microbiota diversity was negatively correlated with ORV immunogenicity, suggesting that high early-life microbial exposure may contribute to impaired vaccine efficacy.

This is a little confusing, suggest modifying to "increased diversity was negatively correlated....

Line 136 – do the authors mean and/or or either/or here for definition of infection?

Figure 2C – sample size for IND neo + appears incorrect (looks like IND total) or the legend should not be neo+

Line 200, add significance for breastmilk IgA negatively correlated with infant post-vaccination RV0IgA in India xxx and Malawixxx"

Line 244, typo – `this

Table S6 - why were only taxa with a prevalence of 20% assessed? suggest to include rationale in methods

Reviewer #1 (Remarks to the Author):

The authors have revised this manuscript extensively in response to the reviewers' comments. My concern were principally with the presentation of data in figures and supplement. I consider that these comments have been fully addressed in the revised manuscript.

Author response: We thank the reviewer for their feedback.

Reviewer #2 (Remarks to the Author):

Parker and Bronowski et al have significantly revised the manuscript and adequately addressed reviewer comments.

This is a rich and complex data set derived from a thoughtfully-designed prospective longitudinal study that has been well analyzed and will have importance to the rotavirus and vaccinology fields and global health.

Some minor comments

Line 42/43 in abstract, In both India and Malawi, pre-vaccination microbiota diversity was negatively correlated with ORV immunogenicity, suggesting that high early-life microbial exposure may contribute to impaired vaccine efficacy.

This is a little confusing, suggest modifying to "increased diversity was negatively correlated...."

Author response: We have modified the wording as follows: *'increased microbiota diversity was negatively correlated...'*

Line 136 – do the authors mean and/or or either/or here for definition of infection?

Author response: We now specify that the definition is being *'and/or'*.

Figure 2C – sample size for IND neo + appears incorrect (looks like IND total) or the legend should not be neo+

Author response: Thank you for spotting this. The reported numbers have been updated to reflect the IND neo+ population.

Line 200, add significance for breastmilk IgA negatively correlated with infant post-vaccination RV0IgA in India xxx and Malawixxx"

Author response: As suggested, we now report Pearson's correlation coefficients for these associations as follows: *'...was negatively correlated with infant post-vaccination RV-IgA levels in India and Malawi (r of -0.14 and -0.26, respectively; Figure 2C).'*

Line 244, typo – 'this

Author response: Thank you for highlighting this. The typo has now been corrected.

Table S6 - why were only taxa with a prevalence of 20% assessed? suggest to include rationale in methods

Author response: This threshold was selected based on prior validation of zero-inflated negative binomial longitudinal models by the NBZIMM package developers. We have clarified this and cited the relevant paper (Zhang & Li; *BMC Bioinformatics* **21**; 488) in the methods section as follows (line 934):

“A threshold of 20% was selected based on prior validation of this statistical modelling approach in the context of microbiome proportion data⁴⁵.”